# Western tropical Pacific multidecadal variability forced by the Atlantic multidecadal oscillation

Cheng Sun[1], Fred Kucharski[2,3], Jianping Li[1,4], Fei-Fei Jin[5], In-Sik Kang[3,6] & Ruiqiang Ding[7]

Observational analysis suggests that the western tropical Pacific (WTP) sea surface temperature (SST) shows predominant variability over multidecadal time scales, which is unlikely to be explained by the Interdecadal Pacific Oscillation. Here we show that this variability is largely explained by the remote Atlantic multidecadal oscillation (AMO). A suite of Atlantic Pacemaker experiments successfully reproduces the WTP multidecadal variability and the AMO–WTP SST connection. The AMO warm SST anomaly generates an atmospheric teleconnection to the North Pacific, which weakens the Aleutian low and subtropical North Pacific westerlies. The wind changes induce a subtropical North Pacific SST warming through wind–evaporation–SST effect, and in response to this warming, the surface winds converge towards the subtropical North Pacific from the tropics, leading to anomalous cyclonic circulation and low pressure over the WTP region. The warm SST anomaly further develops due to the SST–sea level pressure–cloud–longwave radiation positive feedback. Our findings suggest that the Atlantic Ocean acts as a key pacemaker for the western Pacific decadal climate variability.

[1] State Key Laboratory of Earth Surface Processes and Resource Ecology and College of Global Change and Earth System Science, Beijing Normal University, Beijing 100875, China. [2] Earth System Physics Section, Abdus Salam International Centre for Theoretical Physics, Strada Costiera, 11, Trieste 34151, Italy. [3] Center of Excellence for Climate Change Research, Department of Meteorology, King Abdulaziz University, Jeddah 21589, Saudi Arabia. [4] Laboratory for Regional Oceanography and Numerical Modeling, Qingdao National Laboratory for Marine Science and Technology, Qingdao 266237, China. [5] Department of Atmospheric Science, University of Hawaii–Manoa, Honolulu, Hawaii 96822, USA. [6] School of Earth Environment Sciences, Seoul National University, Seoul 151-742, Korea. [7] State Key Laboratory of Numerical Modeling for Atmospheric Sciences and Geophysical Fluid Dynamics (LASG), Institute of Atmospheric Physics, Chinese Academy of Sciences, Beijing 100029, China. Correspondence and requests for materials should be addressed to C.S. (email: scheng@bnu.edu.cn) or to J.L. (email: ljp@bnu.edu.cn).

Global mean sea surface temperature (SST) shows pronounced warming trend for the instrumental period and there is strong evidence that the warming trend is related to the increase in greenhouse gas concentration[1]. Nevertheless, the global SST warming is not spatially uniform and even a long-term cooling trend can be observed in the high-latitude North Atlantic basin[2,3]. In the tropics, there is still large uncertainty with respect to the long-term change in the eastern tropical Pacific SST[2,4,5], while in contrast, the western tropical Pacific (WTP; 0°–25° N, 130°–170° E) shows a significant SST warming trend and there is a widespread consensus about the WTP long-term warming in the literature[2,6,7]. Since the warm WTP SST is a major heat and moisture source for the global atmospheric circulation and has important implications for regional and global climates[8], it is obvious that a better understanding of the change and variability in the WTP SST is necessary.

Overlapping the significant warming trend, strong variability over various time scales are also present in the WTP SST. At interannual time scales, there is a strong simultaneous relationship between the WTP SST and El Niño–Southern Oscillation (ENSO) in winter, and the WTP SST anomalies play an important role in conveying the delayed impact of ENSO on summer climate over East Asia[9–11]. In contrast to the extensive knowledge on the WTP interannual variability, very few studies have been carried out focusing on the WTP decadal (including multidecadal) variability. Recent studies have investigated the WTP sea level multidecadal variability and suggested that the sea level variations may be influenced by the Pacific decadal oscillation[12] and Indian Ocean SSTs[13]. The warming of WTP SST in recent two decades and associated deepening of the thermocline is related to the ocean heat uptake during the global warming hiatus period[14,15]. The decadal variability of WTP SST also has important implications for the decadal variations in extratropical atmospheric circulation[16] and western North Pacific typhoon activity[17]. Nevertheless, the mechanism responsible for the WTP SST decadal fluctuations remains elusive.

Over the interdecadal time scales, the SST variability over Pacific basin is dominated by the Interdecadal Pacific Oscillation (IPO). The IPO is used to describe the ENSO-like pattern that can be obtained as a leading mode of empirical orthogonal function analysis of global detrended low-pass-filtered SST data[18]. Previous studies have suggested that the IPO may be a Pacific-wide manifestation of the Pacific decadal oscillation in the North Pacific and both correspond to the same physical phenomenon[19]. The IPO shows fluctuations over decadal to multidecadal timescales and is significant correlated with a number of climate indices around the Pacific[19]. However, there is almost no simultaneous correlation between the IPO and decadal fluctuations of WTP SST (Supplementary Fig. 1), indicating that the WTP SST decadal variability is unlikely to be explained by the IPO mode.

Another important mode of decadal climate variability is the Atlantic multidecadal oscillation (AMO) in the North Atlantic basin, which is characterized by a spatially coherent pattern with basin-wide warm/cold SST anomalies[20]. There has been strong evidence that changes in ocean heat transport associated with the Atlantic meridional overturning circulation are first order important for AMO[21–23]. The AMO shows an oscillatory behaviour between warm and cold phases with a period of 50–70 years[24,25]. The persistent basin-scale SST anomalies associated with the AMO have significant impacts on the climate not only in and around the North Atlantic[26,27], but also in remote regions[28–32]. Several studies have suggested that there is an Atlantic–Pacific connection at low-frequency time scales,

but they have focused on the North Pacific decadal variability[33], the eastern Pacific response (or IPO-like pattern) to the Atlantic forcing[34,35], and the trend of the tropical atmosphere–ocean coupled system over recent decades[36–38]. It is still unknown whether there is a close relationship between AMO and the decadal fluctuations of WTP SST.

In this study, we perform statistical analyses and a suite of Atlantic Pacemaker experiments to show that the observed WTP multidecadal variability is forced by the AMO. A subtropical North Pacific bridge mechanism is identified to explain the AMO teleconnection to the WTP region, and the SST–sea level pressure–cloud–longwave radiation feedback is found to play an important role in the development of SST anomalies associated with the WTP multidecadal variability. Our results highlight the remote influence of the Atlantic Ocean on western Pacific decadal climate variability.

## Results

**Connection between Atlantic and western tropical Pacific.** Figure 1 shows the normalized time series of annual mean AMO and WTP SST indices (see Methods) for 1900–2013 derived from the Kaplan SST data. The fluctuations in the WTP SST are strongly in phase with those in the AMO over decadal time scales. Highly positive correlation coefficients are observed between the AMO and WTP SST indices at zero lag based on both unfiltered ($r = 0.80$) and 11-year running mean ($r = 0.93$) data, both significant at the 95% confidence level. Since the raw AMO and WTP indices both show a warming trend for 1900–2013 (Fig. 1a), we re-examine the correlation based on the data after removing the linear trend (Fig. 1b), and the positive correlation between detrended AMO and WTP SST remains strong ($r = 0.68$ for unsmoothed data, $r = 0.90$ for 11-year running means, both significant at the 95% confidence level). Moreover, significant simultaneous correlations are also found for other choices of the running window length (Supplementary Fig. 2). Furthermore, the decadal variations in the WTP SST are consistent across different SST data sets (Supplementary Fig. 3), indicating that the results are not sensitive to the exact choice of data set. Power spectral analysis is further applied to detect the predominant timescales of the WTP SST (Fig. 2). The spectrum of detrended WTP SST is characterized by a high and significant concentration of variance at multidecadal time scales with a peak at quasi-60 years (here the shortness of observational record length forms a limitation in the spectral analysis), indicating pronounced multidecadal variability over the WTP region. Thus, the observational analysis suggests that the WTP SST shows very strong multidecadal variability that is significantly correlated with the AMO. The spatial patterns of the AMO–WTP SST relationship over decadal time scales are further analysed. Figure 3 shows the correlation map between 11-year running mean WTP SST index and SSTs north of 20° S for the detrended data, and the correlations over the North Atlantic exhibit a basin-wide uniform pattern with significant positive values that resembles the AMO. Meanwhile, the SST correlation patterns for the WTP decadal variability are consistent among different SST data sets. Thus, these results further suggest that the observed AMO–WTP SST connection is strong and robust.

The WTP SST decadal variability and its connection to the AMO are reasonably well reproduced in the ATL_VARMIX experiment (Atlantic Pacemaker experiment: AGCM coupled to mixed-layer slab ocean model in Indo–Pacific whereas observational SSTs prescribed in Atlantic; see Methods). The simulated WTP SST index also exhibits strong multidecadal variability with a significant spectral peak occurring at around 60 years (Fig. 2), and the amplitude of WTP SST decadal

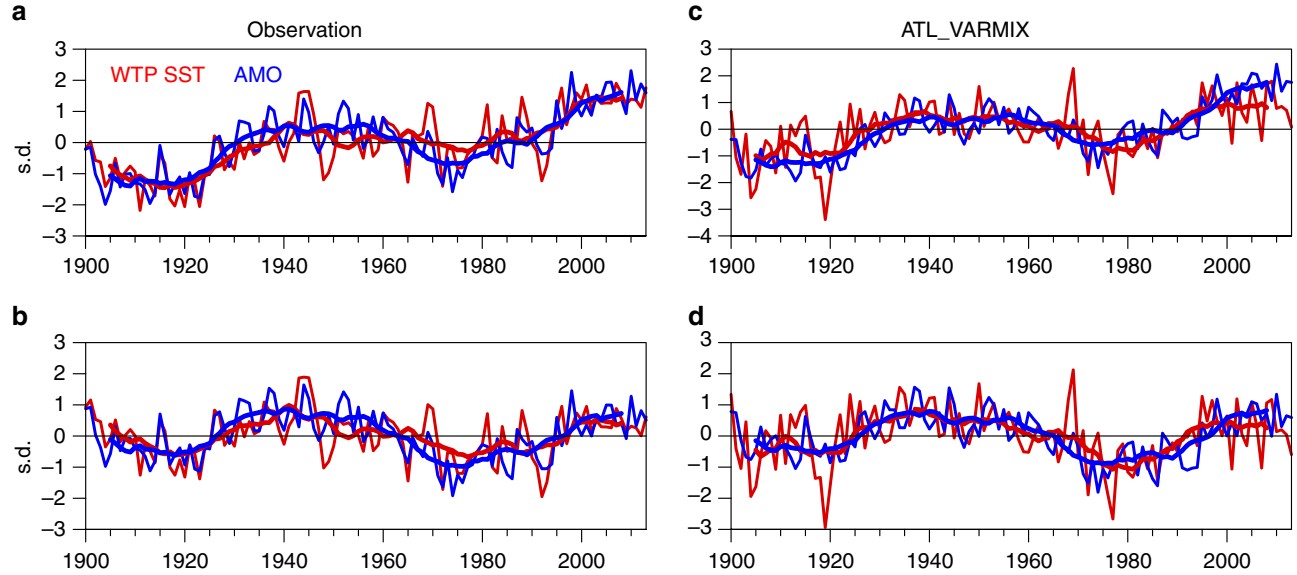

**Figure 1 | The Atlantic–western tropical Pacific SST connection in the observations and ATL_VARMIX simulations.** (**a**) Time series of the western tropical Pacific SST and Atlantic multidecadal oscillation indices for the period 1900–2013 (thin lines) and 11-year running averages (thick lines), normalized by the long-term s.d. (**b**) As in **a**, but for the time series after removing the long-term linear trend. (**c,d**) As in **a** and **b** but based on the data from the ATL_VARMIX simulations.

variability is about 0.1 K, very close to the observations (0.1, 0.1 and 0.12 K for Kaplan, ERSST and HadSST3, respectively). The simulated WTP SST decadal fluctuations are in-phase with the observed WTP SST (Fig. 1), and as expected, the correlation between AMO and simulated WTP SST over decadal time scales is high (0.91 and 0.89 with and without inclusion of the linear trend, respectively, significant at the 95% confidence level), comparable to the observation. The correlation for the unfiltered AMO and WTP SST in the simulation is slightly lower than in the observation (0.56 and 0.43 with and without inclusion of the linear trend, respectively, significant at the 95% confidence level). Although there exist very small differences between the AMO indices in the ATL_VARMIX experiment and the observation (because they are derived from different SST data sets: HadISST1 for the ATL_VARMIX experiment), the decadal fluctuations in both AMO indices are similar. Therefore, in both simulations and observations, the WTP SST decadal variability can be largely explained by the AMO. Meanwhile, in the simulation, high and significant positive correlations extend across almost the entire North Atlantic in the correlation map of the SST field with decadal WTP SST index (Fig. 3d), consistent with the observation (Fig. 3a–c). The only difference is that the eastern Pacific SSTs show significant negative correlations with the decadal WTP SST, while in the observation weak negative correlations are observed only in the eastern equatorial Pacific.

**Thermodynamic processes**. The mechanisms responsible for the WTP decadal variability are further investigated using the ATL_VARMIX simulation, and a surface heat budget analysis is performed by regressions of four heat flux components (see Methods; Fig. 4a–d) and the net heat flux (Supplementary Fig. 4) onto the decadal WTP SST index. Positive (negative) net surface heat flux anomaly over the WTP is associated with the SST warm (cold) phase, and this is consistent with that in the SOM the SST variation is driven by the net heat flux into the ocean. The spatial patterns of net longwave radiation and sensible and latent heat fluxes are consistent with the decadal WTP SST, showing positive anomalies over the WTP region, while the shortwave radiation shows negative anomalies, opposite to the WTP SST. Lagged regressions of the flux components on the

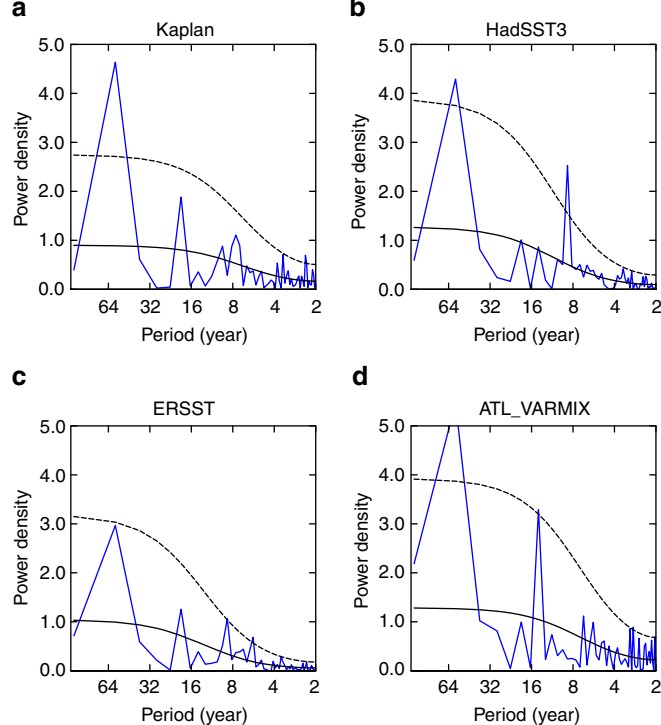

**Figure 2 | Power spectral analysis of the western tropical Pacific SST index.** (**a–d**) Power spectra of the western tropical Pacific SST index (1900–2013) from both observations (Kaplan, HadSST3 and ERSST data sets) and the ATL_VARMIX simulations. The long-term trends in western tropical Pacific SST index are removed before spectral analysis. The solid and dashed black lines in **a–d** indicate the corresponding red noise spectra (solid black line) and 95% confidence levels for power spectra (dashed black line).

decadal WTP SST (Supplementary Fig. 4) further indicate that net longwave radiation and sensible and latent heat fluxes are responsible for the positive net heat flux anomaly into the ocean, which warms up the WTP SST, while the shortwave radiation acts

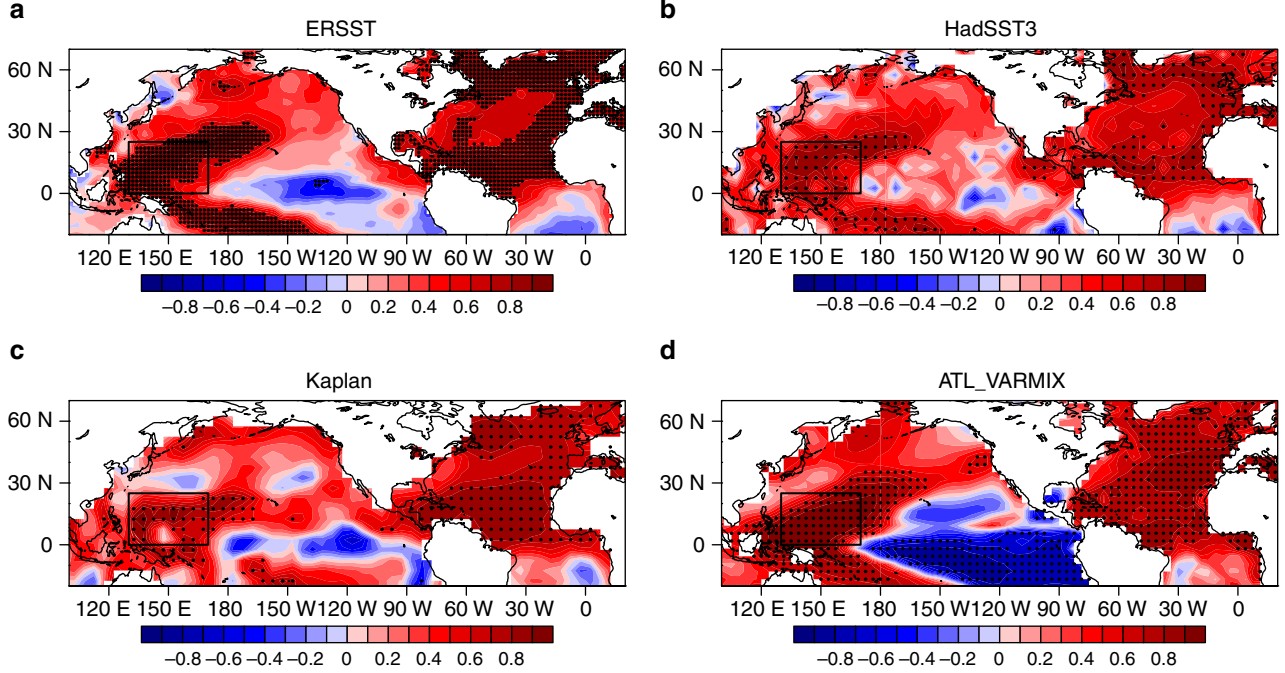

**Figure 3 | Teleconnection between the Atlantic multidecadal oscillation and western tropical Pacific decadal variability.** (**a**) Correlation map between the western tropical Pacific SST index and global SST north of 20° S over decadal timescales for the period 1900–2013 in the ERSST data set. The black box indicates the region used to define the western tropical Pacific SST index (0°–25° N, 130°–170° E). (**b**–**d**) As in **a**, but based on the data from the HadSST3 data, Kaplan SST data and ATL_VARMIX simulations. In **a**–**d**, the long-term linear trends in SST data were removed prior to the correlation analysis, and dots indicate the correlations significant at the 95% confidence level.

as a damping of the WTP SST and counter-acts the other three components. Moreover, the net longwave radiation is the most dominant component forcing the SST decadal variation, and to a less extent the sensible and latent heat fluxes.

The surface heat flux anomalies are closely related to the variations in atmospheric circulation. In the ATL_VARMIX simulation, the sea level pressure (SLP) field (Fig. 4e) associated with the decadal WTP SST exhibits an anomalous low over the WTP region. The anomalous low corresponds to the low-level convergence anomaly and can result in anomalous ascending motion over the WTP. The ascending motion in tropical region often leads to the formation of clouds, especially of convective high-level clouds. As expected, in association with the decadal warm SST anomaly, the cloud amount over the WTP region increases remarkably, and this is largely due to the increase of convective high-level clouds (Fig. 4f). The increase of cloud cover has completely opposite effects on the longwave and shortwave radiation. Increased convection and cloudiness can trap more longwave radiation and lead to less longwave heat loss from the surface, warming up the surface, but it reduces the surface shortwave radiation due to cloud reflection, which tends to damp the SST anomaly. This is consistent with the simulated anomalies of net surface longwave radiation (SLR) and shortwave radiation associated with the decadal WTP SST. Moreover, these results indicate a SST–SLP–cloud–longwave radiation positive feedback over the WTP region; that is the warm SST anomaly can give rise to the anomalous low and leads to the increased cloud cover, which reduces the longwave heat loss from the surface, reinforcing the warm SST anomaly. The positive feedback between SLR and SST has also been revealed in both observational[39,40] and modelling[41,42] studies, and particularly, it has been applied to explain the Atlantic warm pool and its variability[40]. Thus, our analysis provides further modelling evidence that the positive feedback plays a key role in amplifying the WTP

SST decadal variations. In addition, the anomalous low over the WTP corresponds to a cyclonic circulation anomaly which weakens the climatological northeasterly trade winds over the region (Supplementary Fig. 4b), and thus leads to the anomalous turbulent surface heat fluxes from the atmosphere to the ocean.

Besides the cloud feedback, we also investigated the effect of water vapour (WV) feedback on the WTP multidecadal variability using the ATL_VARMIX simulations. At decadal timescales, both cloud cover and column-integrated WV are highly and positively correlated with the net SLR over the WTP region (0.97 and 0.91 for cloud cover and WV, respectively, Supplementary Fig. 5a). To extract the respective influence of cloud cover and WV, we further performed the partial correlation analysis. The partial correlation between the decadal SLR and WV is considerably reduced and becomes insignificant ($r = 0.35$) with the influence of cloud cover excluded, whereas that between the decadal SLR and cloud cover remains strong and significant ($r = 0.89$) after removing the contribution from the WV. Thus, over the WTP region, the decadal variability of SLR is more closely related to the cloud cover than to the WV, and the cloud feedback to the SLR is independent of the WV feedback. Moreover, we established an empirical linear model for the decadal SLR based on the cloud cover and column-integrated WV (Supplementary Fig. 5), and the decadal SLR fitted using the linear model closely follows the simulated SLR ($r = 0.98$). The respective contribution from cloud feedback and WV feedback to the decadal SLR anomalies over the WTP region can be further estimated by using the linear model. In association with the WTP SST multidecadal variability, the contribution from WV feedback to the SLR anomalies is about 0.06 W m$^{-2}$ (Supplementary Fig. 5d), much smaller than that from the cloud feedback (0.31 W m$^{-2}$, Supplementary Fig. 5e). Therefore, in the ATL_VARMIX simulation, the anomalous SLR associated with

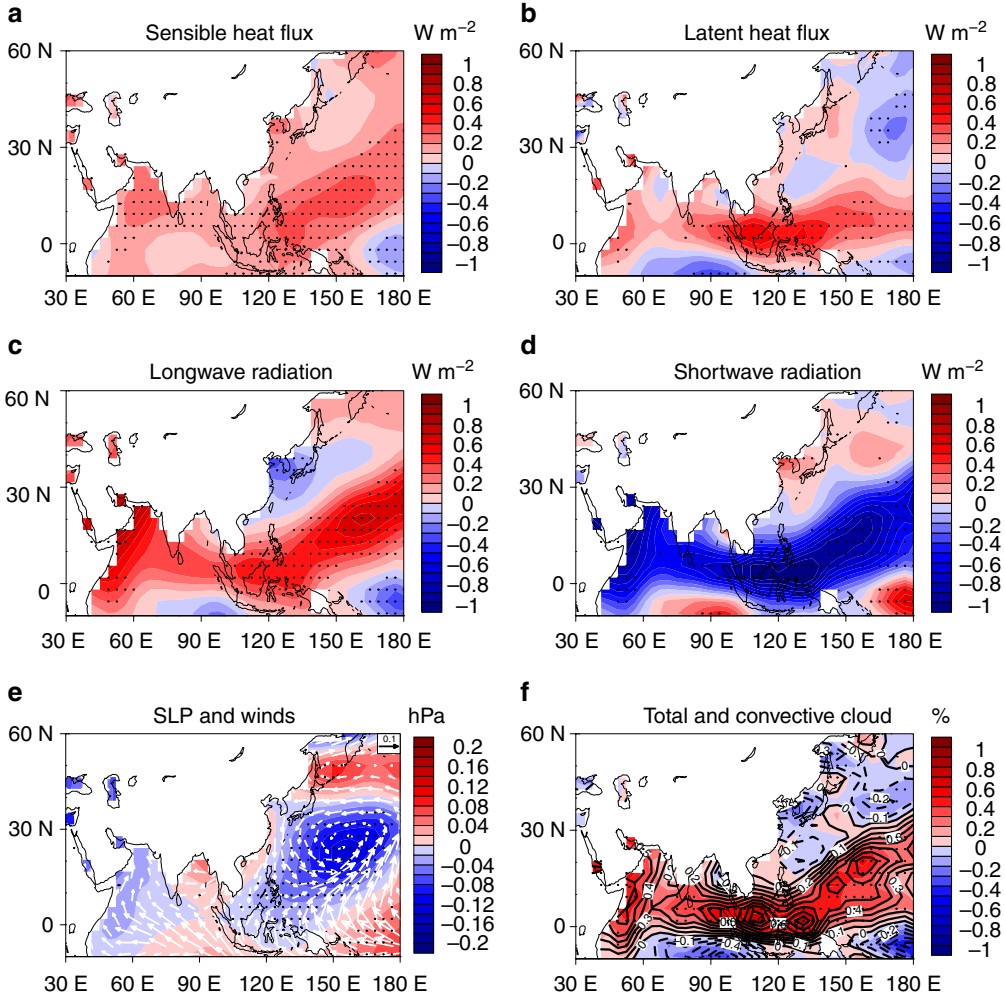

**Figure 4 | Physical processes involved in the western tropical Pacific SST decadal variability in the ATL_VARMIX simulations.** The regression maps of (**a**) surface sensible heat flux, (**b**) surface latent heat flux, (**c**) net surface longwave radiation and (**d**) net surface shortwave radiation (units: W m$^{-2}$) with respect to the normalized western tropical Pacific SST index at decadal time scales. All fluxes are defined to be positive downward. (**e**) Regressions of the sea level pressure (shading, units: hPa) and surface wind filed (vector, units: m s$^{-1}$) with respect to the normalized western tropical Pacific SST index at decadal time scales. (**f**) Regressions of the total cloud amount (shading, units: %) and convective high-level cloud amount (contour, units: %) with respect to the normalized western tropical Pacific SST index at decadal time scales. Dotted shading in **a-f** stands for the regression coefficients significant at the 95% confidence level. The long-term linear trends for 1900–2013 in all variables were removed before the regression analysis.

the WTP multidecadal variability is mainly controlled by the cloud feedback, whereas the WV feedback plays a minor role.

In fact, there is also some observational evidence for the SST–SLP–cloud–longwave radiation positive feedback over the WTP region. The SLP field shows a significant low-pressure signal over the WTP in association with the decadal WTP SST variability (Supplementary Fig. 6), and this anomalous low is observed in both ICOADS and HadSLP products. The cloud amount increases in correspondence to the anomalous low (Supplementary Fig. 6). Due to the shortness of the available records of net SLR, we perform analysis on the raw (unsmoothed) data (Supplementary Fig. 7), ensuring enough degree of freedom to estimate significant correlations. In the records, the variability of SLR over the WTP is consistent in the National Oceanography Centre (NOC) and HOAPS products ($r = 0.75$, significant at the 95% confidence level). This indicates that the variation in the SLR is reliable, since the NOC and HOAPS products are derived from completely different sources. The lagged correlation between SLR and WTP SST anomalies exhibits a nearly symmetrical distribution about the zero lag, with a significant positive correlation

occurring at the zero lag, and the positive correlation is independent of inclusion of the trends in the data. Similar results are obtained in both NOC and HOAPS data sets. Anomalous downward SLR can further enhance the warm SST anomalies. Therefore, Supplementary Figs 6 and 7 together provide observational evidence for the SST–SLP–cloud–longwave radiation positive feedback over the WTP region.

**Subtropical North Pacific bridge mechanism.** The above analysis suggests that the SST–SLP–cloud–longwave radiation positive feedback plays an important role in the WTP SST response to the AMO, but how the positive feedback is excited by the AMO needs to be further addressed. We compare the ATL_VARMIX and ATL_VARAGCM (AGCM forced by observational SST prescribed in Atlantic and climatological SST in Indo–Pacific; see Methods) simulations to explore the possible mechanisms involved. In the ATL_VARMIX simulation, in response to the AMO forcing the tropical upper-level velocity potential shows a zonal dipole structure, with anomalous divergence (convergence) over the

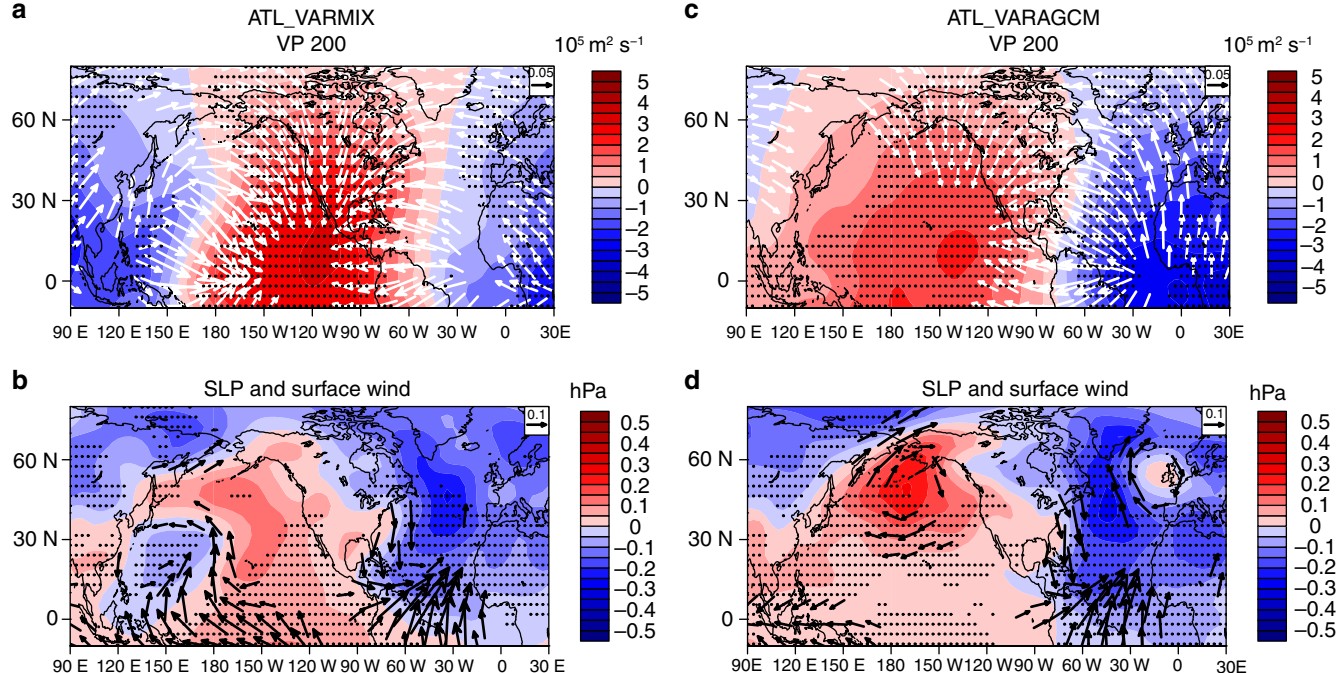

**Figure 5 | Responses of atmospheric circulation to Atlantic multidecadal oscillation SST forcing in the ATL_VARMIX and ATL_VARAGCM.** (**a**) 200 hPa velocity potential (shading, units: $10^5$ m$^2$ s$^{-1}$) and divergent wind (vectors, units: m s$^{-1}$; omitted below 0.05 m s$^{-1}$) anomalies in response to the Atlantic multidecadal oscillation (AMO) forcing in the ATL_VARMIX simulations, shown as regressions onto the normalized AMO index at decadal timescales. (**b**) As in **a**, but for the sea level pressure (shading, units: hPa) and surface winds (vectors, units: m s$^{-1}$; omitted below 0.1 m s$^{-1}$). (**c,d**) As in **a** and **b**, but based on the data from the ATL_VARAGCM simulations. Dotted shading in **a–d** stands for the regression coefficients significant at the 95% confidence level. The long-term linear trends for 1900–2013 in all variables were removed before the regression analysis.

western Pacific (eastern Pacific) region (Fig. 5a), and low-level circulation shows a clear convergence (divergence) signal with anomalous low (high) pressure over the western Pacific (eastern Pacific) region, opposite to the upper level (Fig. 5b). This indicates that anomalous ascending (descending) motion occurs over the western Pacific (eastern Pacific) region. Anomalous ascending motion and surface low pressure is also seen over the tropical North Atlantic from the simulated upper-level divergence and SLP fields, in agreement with a recent observational based analysis[43], and this is due to the heating effect of the AMO warm SST anomaly. Consistent with simulation results found here, there has also been some observational evidence in the literature that an anomalous low (high) and ascending (descending) motions tend to occur over western Pacific when the Atlantic is warmer (colder) than normal, as manifested by the SLP field[35,43,44] and the zonal overturning circulation[37]. The anomalies of SLP and vertical motion are closely related to the change in cloud cover over the WTP, and thus can lead to the occurrence of the SST–SLP–cloud–longwave radiation positive feedback.

A key question is how the anomalous low over WTP is generated in response to the AMO warm SST forcing: is it a direct atmospheric response to the AMO? To address this, we first examine the results from the ATL_VARAGCM simulation (see Methods), which can reflect the atmospheric response directly forced by the AMO. In the ATL_VARAGCM simulation, the upper-level velocity potential and SLP fields shows strong divergence anomaly and anomalous low pressure over the tropical Atlantic, corresponding to the anomalous rising motion generated by the warm AMO SST anomaly. This local atmospheric response to the AMO is generally consistent with the ATL_VARMIX simulation. However, the remote atmospheric response over Pacific basin shows remarkable differences from the ATL_VARMIX. Anomalous upper-level convergence and

surface high pressure are present over the entire Pacific basin, and a strong centre of anomalous high is seen over northern North Pacific. These remote atmospheric responses can be physically interpreted. As shown in Fig. 5c,d, the AMO warm SST anomaly leads to strong ascending motion and upper-level divergence in North Atlantic, with the outflow heading westward and converging over North Pacific, which further induces the compensating subsidence in North Pacific. Thus, the anomalous upper-level divergence over North Atlantic generated by warm SST anomalies is largely compensated by anomalous upper-level convergence in North Pacific, leading to subsidence and high pressure anomalies there. An explanation for the strong high anomaly centre over extratropical North Pacific is that the upper-level convergence in the tropical central-eastern Pacific induced by the North Atlantic warming (Fig. 5c) can generate a Rossby wave train propagating to the extratropics[45,46], enhancing the surface high pressure anomaly caused by the compensating subsidence. In addition, over the WTP region, the anomalous low seen in the ATL_VARMIX simulation is absent in the ATL_VARAGCM. Since the ATL_VARMIX and ATL_VARAGCM simulations only differ in the inclusion of local atmosphere–ocean coupling over basins outside the Atlantic, the above results suggest that the local atmosphere–ocean coupling over the Pacific basin may be important for understanding the connection between AMO and the WTP anomalous low.

The anomalous high pressure over North Pacific in response to the warm AMO phase may influence the SST through wind-evaporation-SST (WES) mechanism[47]. As shown in Fig. 5d, the anomalous high and associated anticyclonic flow can weaken the Aleutian low and decrease the wind speed of subtropical North Pacific (SNP) westerlies. Through the WES effect, the SST over the SNP should become warmer in response to the wind

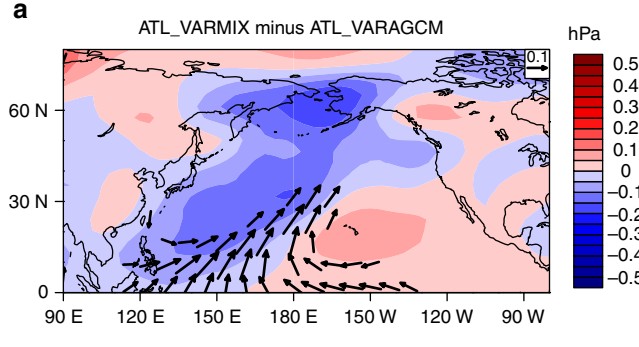

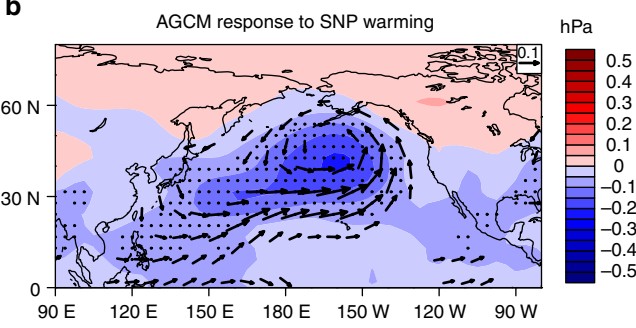

**Figure 6 | The role of the subtropical North Pacific SST anomalies.**
(**a**) The differences in the responses of sea level pressure (shading, units: hPa) and surface winds (vectors, units: m s$^{-1}$; omitted below 0.1 m s$^{-1}$) to the Atlantic multidecadal oscillation (AMO) between ATL_VARMIX and ATL_VARAGCM simulations. The responses to the AMO are obtained by regressions onto the normalized AMO index at decadal timescales. (**b**) Simulated Pacific atmospheric circulation responses of sea level pressure (shading, units: hPa) and associated surface winds (vectors, units: m s$^{-1}$; omitted below 0.1 m s$^{-1}$) to warm SST anomalies over the subtropical North Pacific in the sensitivity experiments using the ICTPAGCM model. Dotted shading in **b** indicates the regions where the results from the sensitivity simulation are significantly different from the control simulation at the 95% confidence level (Student's t-test).

speedy reduction, and thus a positive correlation between SNP SST and AMO should be expected. In fact, in the observations, there are indeed significant warm (cold) SST anomalies over the SNP in association with the warm (cold) AMO phase (Supplementary Fig. 8). Moreover, the SNP SST also exhibits significant covariability with the WTP SST decadal fluctuations in both observations and ATL_VARMIX simulation (Fig. 3). The time series of the SNP decadal SST variations in the simulation and observation data sets are shown in Supplementary Fig. 9a, which both display evident multidecadal fluctuations and vary strongly in phase with each other. Meanwhile, in both the simulation and the observation, the cross correlations of the SNP decadal SST anomalies with the AMO and WTP multidecadal variability (Supplementary Fig. 9b–d) show high and significant simultaneous correlations (above 0.8), indicating that the relationship of the SNP SST with the AMO and WTP SST at decadal timescales is largely in phase (in the observation the positive correlation between the AMO and SNP SST is not only simultaneous but is also significant and slightly larger when the AMO leads by up to several years). Therefore, it appears a reasonable hypothesis that the SNP SST may play a crucial role bridging the AMO and WTP multidecadal variability: the AMO warm/cold SST anomaly induces warmer-than-normal/ cooler-than-normal SST over SNP through WES effect and the SNP SST anomalies in turn exert a feedback on the Pacific

atmospheric circulation, leading to the WTP multidecadal variability.

To further investigate the role of SNP SST feedback and test the above hypothesis, we carried out a sensitivity experiment using the ICTPAGCM (see Methods) to isolate the effect of SNP SST forcing. Figure 6 shows the simulated atmospheric circulation responses from the sensitivity experiment of SNP SST forcing (Fig. 6b) along with the ATL_VARMIX minus ATL_VARAGCM responses (Fig. 6a) for comparison. The difference in Pacific atmospheric responses (for example, SLP and surface winds) to the AMO between ATL_VARMIX and ATL_VARAGCM simulations largely represents the effects of the local SST feedback. As seen in Fig. 6a, the local SST feedback over Pacific during the warm AMO phase leads to anomalous low pressures over WTP and North Pacific and an anomalous high over East Pacific. The associated surface wind is characterized by strong flows converging towards the SNP region with cyclonic and anticyclonic flows over WTP and central-eastern Pacific, respectively. Similar to this atmospheric circulation pattern, the AMO-induced SNP SST warming generates strong anomalous flows converging towards the warm SST anomaly region, leading to an anomalous low and cyclonic circulation over North Pacific. Corresponding to the wind convergence towards the SNP from the tropics, the WTP region is dominated by an anomalous cyclonic flow and low-pressure anomalies (Fig. 6b). Therefore, the different atmospheric circulation responses between ATL_VARMIX and ATL_VARAGCM can be largely explained by the flow pattern generated by the AMO-induced SNP SST warming. In particular, the anomalous low over the WTP region occurring during the warm AMO phase can be explained by the responses to the SNP SST warming. Following the formation of the anomalous low, the SST–SLP–cloud–longwave radiation positive feedback can further develop, leading to the significant SST anomaly over the WTP region.

Meanwhile, in the ATL_VARMIX_SNPCLIM simulation (similar to the ATL_VARMIX experiment, but excluding the effects of SNP SST anomalies; see Methods), the correlations of simulated WTP decadal SST anomalies with the AMO become rather weak and insignificant (Supplementary Fig. 10a, correlation decreases from 0.90 to 0.41, with explained variance reduced by 64%), and the amplitude of AMO-related decadal SST variability over the WTP region is considerably reduced (Supplementary Fig. 10b, amplitude decreases from 0.09 to 0.024 K, reduced by 73%). In addition, without the SNP SST feedback, the Pacific atmospheric circulation response is also different from the result in the ATL_VARMIX. The significant anomalous low and cyclonic flow over the WTP region are absent in the ATL_VARMIX_ SNPCLIM, and the converging flow from the tropics towards the SNP region also disappears, due to the exclusion of SNP SST feedback. Therefore, the bridging role of SNP SST in the connection between AMO and WTP decadal variability is further verified.

Figure 7 finally shows a schematic diagram summarizing the physical mechanisms in the connection between AMO and WTP SST decadal variability. The AMO warm SST anomaly generates anomalous ascent and upper-level divergence over North Atlantic. The associated upper-level outflows converge towards North Pacific, leading to compensating subsidence along with an anomalous high there, which weakens the Aleutian low and decreases the wind speed of SNP westerlies. The wind changes induce a SNP SST warming through WES effect, and in response to this warming, the surface winds converge towards the SNP SST warming centre from the tropics, leading to an anomalous cyclonic circulation and low pressure over the WTP region. The AMO teleconnection to the WTP anomalous low further develops into to a SST warming

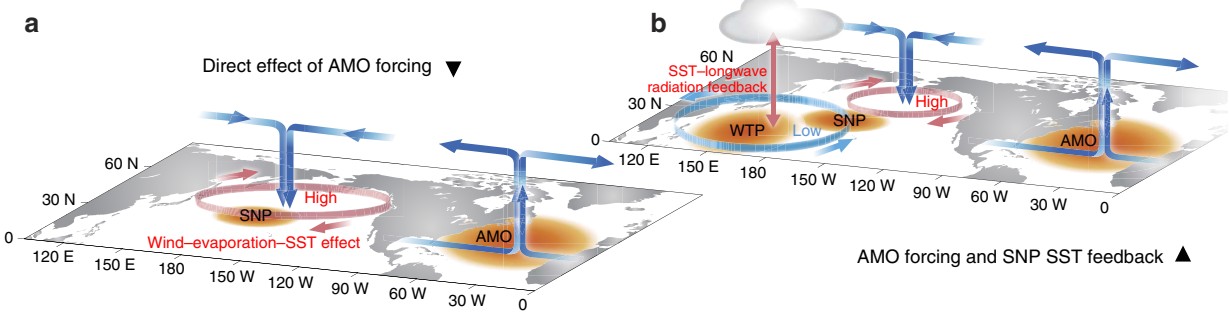

**Figure 7 | Schematic diagram for the teleconnection between Atlantic multidecadal oscillation and western tropical Pacific decadal variability.** (**a**) The direct effect of Atlantic multidecadal oscillation (AMO) forcing on North Pacific. The AMO warm SST anomaly (the orange patch over North Atlantic) generates anomalous ascent and upper-level divergence over North Atlantic. The associated upper-level outflows converge towards North Pacific, leading to compensating subsidence along with an anomalous high there, which weakens the Aleutian low and decreases the wind speed of subtropical North Pacific (SNP) westerlies. The wind changes induce a SNP SST warming (the orange patch over subtropical North Pacific) through wind–evaporation–SST effect. (**b**) The combined effects of AMO and SNP SST warming feedback. In response to the SNP SST warming, the surface winds converge towards the SNP SST warming centre from the tropics, leading to an anomalous cyclonic circulation and low pressure over the western tropical Pacific region. The teleconnection further develops into to a SST warming pattern (the orange patch over western tropical Pacific) due to the SST–sea level pressure–cloud– longwave radiation positive feedback (shown as 'SST–longwave radiation feedback' in the figure).

pattern due to the SST–SLP–cloud–longwave radiation positive feedback.

## Discussion

The WTP ($0°–25°$ N, $130°–170°$ E), a part of the western Pacific warm pool, is a major heat and moisture source for the general atmospheric circulation due to the high SST. Observational analysis suggests that the WTP SST shows predominant variability over multidecadal time scales, which is largely explained by the remote AMO. The mechanisms are investigated using a suite of Atlantic Pacemaker experiments, which successfully reproduces the WTP SST multidecadal variability and the AMO–WTP SST connection. The SNP SST anomalies are identified as a key bridge linking the AMO and WTP SST multidecadal fluctuations. The AMO SST anomalies induce an atmospheric teleconnection from North Atlantic into North Pacific, influencing the SNP SST through the wind–evaporation– SST effect. The AMO-induced SNP SST anomalies provide a feedback to the Pacific atmospheric circulation, which eventually generates SST anomalies over the WTP region via a SST–sea level pressure–cloud–longwave radiation feedback.

Our present analysis shows that the connection between AMO and WTP SST is most pronounced for the multidecadal time scales. A positive correlation is also observed for the unfiltered SST indices in both observations and the ATL_VARMIX simulation, indicating that a potential linkage exists over the interannual time scales and the mechanisms proposed may also be at work. In fact, some previous studies have also suggested that the tropical Atlantic interannual variability, which is highly correlated with the North Atlantic SSTs[48], is related to the ENSO phenomena and the WTP SST anomaly[35,49]. Nevertheless, the interannual connection revealed in this study is less pronounced than the decadal one, particularly in the simulations. A possible explanation is that over the interannual time scales, the WTP SST may also be affected by other variability or noise (for instance, the local stochastic atmospheric forcing and associated surface turbulent fluxes), which reduces the variance explained by the remote influence of AMO. When a decadal low-pass filtering is applied, the interannual forcings are filtered out and the influence of AMO becomes predominant. In addition, a limitation of our study is that the model we used is an AGCM coupled to a mixed-layer ocean model, rather than to a dynamical ocean

model, which considers the role of ocean circulation. We performed a preliminary analysis using the preindustrial control simulations from the CMIP5 fully coupled models, and 2 out of the 18 analysed models (providing a simulation of at least 500 years in length) reproduce the mechanisms suggested in this study. The WTP SST decadal variability and its magnitude are reasonably well reproduced in the mixed-layer ocean model, indicating that the thermodynamics may play the essential role. Despite of this, further modelling studies are required to confirm the relative roles of dynamical and thermodynamic processes in the WTP SST decadal variability.

## Methods

**Data.** The global observational SST data sets used in this study include the Kaplan SST data set[50], the Hadley Centre SST data set (HadSST3)[51] and the extended reconstruction SST version 3 (ERSST v3b) data set[52]. Long-term ship-observed SLP and marine cloud cover were used from the International Comprehensive Ocean–Atmosphere Data Set (ICOADS) release 2.5 (ref. 53), which have been trimmed using a quality control that identifies potential outliers based on the climatological 3.5 s.d. limits. The ICOADS data set is available at the Research Data Archive at the National Center for Atmospheric Research. The SLP field derived from the Hadley SLP data set (HadSLP2)[54] was also employed to test the reliability of the results. The net SLR data were obtained from the NOC Version 2.0 surface flux data set[55] and the Hamburg Ocean Atmosphere Parameters and fluxes from Satellites (HOAPS 3.2) data set[56]. The NOC data set is based on the ship observations from the ICOADS data set and available for the period 1973–2006. The HOAPS data set is fully derived from satellite measurements starting from 1988 and data over the period 1988–2006 are used here. The net SLR is defined to be positive downward in this study.

Because uncertainties in surface observations prior to 1900 are relatively large and the data before 1900 are deemed less reliable[57], we confine our analysis to the post-1900 period for the data sets. We remove the long-term linear trend in the variables for the post-1900 period using the least squares method, and our intent is to remove the centennial scale trends to better isolate and highlight the signal of decadal to multidecadal variability.

The AMO index is defined as the area-weighted average of SST anomalies over the North Atlantic region ($0°–60°$ N, $80°$ W$–0°$) with respect to the 1961–1990 climatology, and an index to describe the WTP SST variability is defined in a similar manner but over the WTP region ($0°–25°$ N, $130°–170°$ E).

**Statistical significance test.** The statistical significance of the linear regression coefficient and correlation between two autocorrelated time series is accessed via a two-tailed Student's $t$-test using the effective number of degrees of freedom $N^{\text{eff}}$, which is given by the following approximation:

$$\frac{1}{N^{\text{eff}}} \approx \frac{1}{N} + \frac{2}{N}\sum_{j=1}^{N}\frac{N-j}{N}\rho_{XX}(j)\rho_{YY}(j),\qquad(1)$$

where $N$ is the sample size and $\rho_{XX}(j)$ and $\rho_{YY}(j)$ are the autocorrelations of two sampled time series $X$ and $Y$ at time lag $j$ (ref. 58).

**Model and experiments.** The model used here is the International Centre for Theoretical Physics AGCM (ICTPAGCM, version 41)[35], coupled to a slab ocean thermodynamic mixed-layer model (SOM). The ICTPAGCM is an intermediate complexity model that contains eight vertical levels with a horizontal resolution of T30. In the SOM, the depth of the mixed layer is constant throughout the whole simulation period varying from 60 m in the extratropics to 40 m in the tropics. The variation of the mixed-layer temperature is derived from the integration of the net heat flux into the ocean (that is, the sum of surface shortwave and longwave radiation, and sensible and latent heat flux; all fluxes are defined to be positive downward).

To examine the effects of Atlantic SST variability on the atmospheric circulation and on SSTs over other ocean basins, we carried out two experiments. The first experiment is referred to as ATL_VARMIX, in which the ICTPAGCM is run coupled with the SOM in the Indo–Pacific region, but observed monthly varying SSTs from HadISST1 product are prescribed in the Atlantic basin (Atlantic Pacemaker experiment)[35]. In the SOM, the mixed layer is used everywhere in the Indo-Pacific region, including both tropics and extratropics, and in the sea-ice region, the surface temperature anomalies are calculated using a slab layer over ice. The transition between the SOM and prescribed SSTs is accomplished by using a buffer zone with a spatial range of 7.5°, in which the SST is calculated from the weighted average of the modelled and prescribed SSTs. The weighting values are one in the prescribed-SST domain and linearly reduced to zero in the buffer zone. The model integrations start in 1872 and run through 2013. An ensemble of five members is generated by restarting the model using small initial perturbations. The first 28 years of all simulations are considered as spin-up and the analysis is performed on the remaining period from 1900 to 2013. The ATL_VARMIX experiment considers both the forcing effect of Atlantic SST variability and atmosphere–ocean coupling in the Indo-Pacific region. In addition, a similar experiment but with climatological monthly SSTs prescribed in the Indo-Pacific region (no coupling) is performed in order to investigate the direct impacts of the Atlantic forcing on the Indo-Pacific atmospheric circulation without inclusion of the feedback from the ocean. An ensemble of five members is performed and this experiment is referred to as ATL_VARAGCM. The results of the five ensemble members from ATL_VARMIX and ATL_VARAGCM experiments were averaged and analysed.

We carried out two experiments to verify the role of SNP SST anomalies as a key bridge linking the AMO and WTP SST anomalies. One is a sensitivity experiment using the ICTPAGCM to examine the response of Pacific atmospheric circulation to the SNP SST forcing. The control run was integrated for 15 years and forced with the climatological SST, and the last 10 years of the integration were used to provide the basic annual mean state. The sensitivity experiment was integrated for 12 years and forced with the warm SNP SST anomaly imposed on the climatological SST. The SNP SST forcing used were five times the regressed SST anomalies onto the AMO index over the SNP regional box (20° N–35° N, 170° E–155° W, as seen in Supplementary Fig. 8). The scaling factor of five was used to make sure that the model produces steady and strong enough responses to the SST forcing with short integrations[59]. A 10-member ensemble mean was constructed from the last 10 years of the integration to reduce the uncertainties arising from different initial conditions. The atmospheric response to the SST forcing is defined as the difference between the ensemble means of the sensitivity and control runs scaled by a factor of 1/5 (ref. 59). The other experiment is similar to the ATL_VARMIX, but with prescribed climatological SST over the SNP region. This experiment aims to investigate whether the WTP decadal SST anomalies can still be influenced by the AMO without the SNP SST feedback, and is referred to as ATL_VARMIX_SNPCLIM.

**Code availability.** The code of ICTPAGCM is available through the URL: https://www.ictp.it/research/esp/models/speedy.aspx.

**Data availability.** The original observational data are publicly available and can be downloaded from the corresponding websites (ERSST, Kaplan, HadSST3, HadSLP2 and ICOADS) at https://www.esrl.noaa.gov/psd/data/gridded/data.noaa.ersst.v3.html, https://www.esrl.noaa.gov/psd/data/gridded/data.kaplan_sst.html, http://www.metoffice.gov.uk/hadobs/hadsst3/, http://www.metoffice.gov.uk/hadobs/hadslp2/ and ftp://ftp.cdc.noaa.gov/Datasets/icoads2.5/. The authors declare that the data supporting the findings of this study are available upon request.

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

## Acknowledgements

This work was jointly supported by the National Key Research and Development Plan (2016YFA0601801) and the National Programme on Global Change and Air–Sea Interaction (GASI-IPOVAI-06 and GASI-IPOVAI-03). C.S. conducted parts of the study during research stays at the International Centre for Theoretical Physics, Trieste, Italy through the Junior Associateship Scheme.

## Author contributions

C.S., F.K. and J.L. designed the research. C.S. performed the data analysis, prepared all figures and led the writing of the manuscript. All the authors discussed the results and commented on the manuscript.

## Additional information

**Competing interests:** The authors declare no competing financial interests.

