## [Peer Review File Description · Nature Communications]

Reviewers' comments:

Reviewer #1 (Remarks to the Author):

Review of the manuscript "Western tropical Pacific multidecadal variability forced by the Atlantic multidecadal oscillation" (NCOMMS-17-00941) written by Dr Sun and colleagues, submitted to Nature Communications.

The manuscript presents very interesting analyses of the impact of AMO on western tropical Pacific (WTP) multidecadal SST variability, using both observations and a suite of numerical simulations. The results suggest that the observed WTP multidecadal SST variability is mainly induced by the AMO through atmospheric teleconnections. This study is an important contribution to the field. In particular, the successfully reproduced WTP multidecadal variability in the Atlantic Pacemaker experiments is novel and has not been shown before. The results presented in the manuscript are of immediate interest to many people in the research areas of climate variability and change, as well as to people from a broader range of areas such as the economic impact of climate changes. Due to the above novel results and broad impacts, I recommend the manuscript be accepted for publication in Nature Communications after some minor revision outlined in the following specific review comments.

1, What are the cross correlations between SNP and WTP decadal SST anomalies at all time leads/lags? i.e. are they in phase or there is some time lead/lag between them?

2, Similarly, what are the cross correlations between SNP decadal SST anomalies and the AMO at all time leads/lags? i.e. are they in phase or there is some time lead/lag between them?

3, Since SNP SST anomalies are identified as a key bridge linking the AMO and WTP SST anomalies, if possible it would be nice to have an experiment as ATL_VARMIX but with prescribed climatology SST over the SNP region, i.e. if there are no SNO SST anomalies, can WTP decadal SST anomalies still be influenced by the AMO ?

4, A very recent observational based analysis (Barcikowska et al. 2017) on the teleconnection of multidecadal variability also shows that warmer WTP SST and negative SLP anomalies occur during the positive AMO phase, and vice versa (Fig. 2 A,C), and discussed that "Changes of SSTs in the tropical and extratropical North Atlantic are mirrored by changes throughout the western Pacific and Indian Oceans". Fig. 4 in Barcikowska et al. 2017 shows the regression of the observed velocity potential at 850hPa and 200hPa on the AMO, and discussed that "Warming observed in the tropical North Atlantic is physically consistent with the anomalous large-scale low-level convergence and high-level divergence tendencies, indicative of strengthened convection in this region. Cooling in the tropical central Pacific is associated with a relative subsidence tendency or suppressed convection tendencies in that region", quite consistent with the proposed mechanism (Fig. 3 and Fig. 5) in the manuscript. It would be very useful to cite this study to support the conclusion of this manuscript.

Barcikowska, M.J., T. R. Knutson, R. Zhang, 2017, Observed and Simulated Fingerprints of Multidecadal Climate Variability and Their Contributions to Periods of Global SST Stagnation, *Journal of Climate*, 30, 721-737, doi: <http://dx.doi.org/10.1175/JCLI-D-16-0443.1>

5, Supplementary Fig. 1: it would be nice to show the cross-correlation between low-pass filtered WTP SST and IPO/PDO at other time lags as well, so the time lead/lags at the maximum correlations can be seen.

6, Supplementary Fig. 4 and 5 are very important results, and I would suggest to move them to the main manuscript. In particular, Supplementary Fig. 5 shows the spatial WTP SST pattern associated with the AMO, and is very useful to understand the discussions in the main text.

7, Why the color bar ranges in Fig. 2f and Supplementary Fig. 7b are very different? Both are regressions of cloud cover on the decadal WTP SST index.

8, Line 196, suggest changing the word "driving" to "amplifying", since here it refers to a positive feedback, not a driving mechanism.

Reviewer #2 (Remarks to the Author):

The study investigates the mechanisms behind the western tropical Pacific (WTP) SST multidecadal variability. The manuscript claims that the multidecadal variability of the WTP is driven by the Atlantic Multidecadal Oscillation (AMO). The teleconnection process occurs in two steps:

1. the positive phase of the AMO generates upward motion over the North Atlantic that descends over the remote North Pacific, weakening the Aleutian low and reducing wind speed in the subtropical North Pacific westerlies. The weakening of the westerly winds in turn lead to a wind- evaporation-SST effect that warms the SST locally.

2. The subtropical North Pacific SST warming induces wind convergence from the tropics, which drives an anomalous cyclonic circulation over the WTP. The anomalous atmospheric circulation eventually generates SST warming over the region via a SST - sea level pressure – cloud – longwave radiation feedback.

The mechanism is very complex, but the authors provided enough evidence of the results. The findings are interesting, novel and worth publication.

One thing that the authors could have considered in this study is the use of a fully-dynamical ocean model, instead of a partially coupled mixed ocean model. This is because the simulation suggests a link between the AMO and the eastern Pacific as shown by the strong and significant correlations in Supplementary Fig.5d. Thus, there could exist a dynamical link across the ocean basin. Having said that, I understand that performing a Pacemaker experiment in such configuration may not be feasible for this study. But I wonder if this mechanism could be simulated in another model (e.g. pre-industrial control run from any cmip model that can reproduce a reasonable AMO). The authors correctly acknowledged the model design limitation at the end of the manuscript.

Line 199 seems to have the wrong figure reference. Should that be Figure 2e or Supplementary Figure 6b?

Model description: The ICTPAGCM is coupled to a slab ocean model in the Indo-Pacific region and monthly varying SSTs in the Atlantic. The model design needs a clearer description, i.e. what is the forcing in the extratropical regions (e.g. southern latitudes)? How is treated the transition between slab ocean model and prescribed SST?

Figure 1b: for comparison purposes it may be better to scale the y-axis to the same as the other subpanels.

Suppl. Fig. 3b: as above, it may look better if y-axis range is increased.

Reviewer #3 (Remarks to the Author):

Review's comments for the paper (NCOMMS-17-00941), entitled "Western tropical Pacific multidecadal variability forced by the Atlantic multidecadal oscillation" by Sun and co-authors

Recommendation: Major revision

This paper investigate the multidecadal sea surface temperature (SST) variability over western tropical Pacific (WTP). By using both observational data and numerical experiments, the paper suggests that the WTP multidecadal variability is largely explained by the remote Atlantic multidecadal oscillation (AMO). The paper further argues that the SST–sea level pressure (SLP)–cloud–longwave radiation positive feedback is the main physical process involved in the response of WTP SST change to AMO change. Results are interesting and they are generally well presented. Therefore, the paper is worth of publication. However, the paper needs some improvements by properly address the comments listed below before it can be accepted for publication in the Nature Communications.

Major comments

1. Paper suggests that SST–sea level pressure (SLP)–cloud–longwave radiation positive feedback is the main physical process involved in the WTP SST response to AMO. With SST warming, it is expecting an increase in atmospheric water vapor through local enhanced evaporation. What is the role of water vapor feedback? The reviewer strongly recommends authors to investigate this issue, and quantify its role, at least in the pacemaker experiments.
2. The study proposes that the change in SST over subtropical North Pacific (SNP) bridges the influence of AMO on the WTP multidecadal variability. However, in both observations and model experiments, the time series of SST variability over SNP is not analysed. To make your arguments more convincing, this time series needs to be analysed and its phase relationships with both AMO and the WTP multidecadal variability should be discussed.
3. Conclusion and discussion could be more concise by focusing on main points. There is no need to repeat some statements that are already in results section.

Minor comments

1. Lines 62-65. Need to make a distinction between simultaneous relationship in winter and delayed impact of ENSO on summer climate over East Asia.
2. Lines 93-95. There are many studies about AMO remote impacts, but the authors only cite their own two papers published very recently. Suggest to cite some of those early studies.
3. Lines 118-121. These power spectra are based on about 140 year data. Need some comments on limitation.
4. Supplementary figure 5d indicates pattern of negative phase of IPO. What is the relationship between AMO and IPO (PDO) indices in the model experiment? Does IPO in the model experiments play a role in the WTP multidecadal variability?
5. Lines 290-295. The information about these sensitive experiments should be in the method section.

I. Response to Comments of Reviewer #1

General comments:

Review of the manuscript “Western tropical Pacific multidecadal variability forced by the Atlantic multidecadal oscillation” (NCOMMS-17-00941) written by Dr Sun and colleagues, submitted to Nature Communications.

The manuscript presents very interesting analyses of the impact of AMO on western tropical pacific (WTP) multidecadal SST variability, using both observations and a suite of numerical simulations. The results suggest that the observed WTP multidecadal SST variability is mainly induced by the AMO through atmospheric teleconnections. This study is an important contribution to the field. In particular, the successfully reproduced WTP multidecadal variability in the Atlantic Pacemaker experiments is novel and has not been shown before. The results presented in the manuscript are of immediate interest to many people in the research areas of climate variability and change, as well as to people from a broader range of areas such as the economic impact of climate changes. Due to the above novel results and broad impacts, I recommend the manuscript be accepted for publication in Nature Communications after some minor revision outlined in the following specific review comments.

Response: We are very grateful for the reviewer’s helpful comments, which improve the manuscript significantly. We have made changes to the manuscript carefully, according to the reviewer’s comments and suggestions. The point-to-point responses to the comments are listed as follows.

Specific comments:

1, What are the cross correlations between SNP and WTP decadal SST anomalies at all time leads/lags? i.e. are they in phase or there is some time lead/lag between them?

Response: We thank the reviewer for the enlightenment and good comments. We performed additional analyses in light of the reviewer’s questions, and investigated the cross correlations between SNP and WTP decadal SST anomalies at all time leads/lags in both the observation and ATL_VARMIX simulation. The time series of the SNP decadal SST variations in the simulation and observation datasets are shown in Fig. A1a, which both display evident multidecadal fluctuations and vary strongly in phase with each other. This indicates that the observed SNP decadal SST variability is reasonably reproduced by the ATL_VARMIX simulation. The cross correlations between the SNP and WTP decadal SST anomalies in both the simulation and the observation are further shown in Fig. A1b and A1d (the blue line) as a function of the time lag. Since the results from the two SST observation datasets are similar, here we show the observational based analysis from the ERSST data. The cross correlations

between the observed SNP and WTP decadal SST anomalies are generally consistent with the simulated results, showing a high and significant correlation peak around lag zero. Although the maximum correlation occurs at lag 1 for the observation, it is still reasonable to consider the simultaneous relationship since the multidecadal time scale dominated in the two indices is much broader than the lag. The simultaneous correlations are 0.85 and 0.86 for the observation and the simulation, respectively. This suggests that at the decadal timescales, the WTP and SNP SST anomalies are largely in phase: warm (cool) WTP SST anomalies are likely to occur when there is an anomalous SST warming (cooling) over the SNP region.

To further confirm the simultaneous relationship between the WTP and SNP SST anomalies in the observation, we also analyzed the cross correlation using the unfiltered data. This analysis can also be used to test whether the 1-year lag really exists in the annually resolved time series. As shown in Fig. A1c, the maximum correlation appears at lag zero with a clear correlation peak, and the correlations quickly diminish with time lag (the 1-year lag vanishes). Therefore, this provides further evidence that the WTP and SNP SST anomalies vary largely in phase with each other. According to the reviewer’s comments, we include the results for the relationship between SNP and WTP decadal SST anomalies in the revised manuscript (Supplementary Fig. 9 and lines 301-310).

Figure A1 Phase relationship of the SNP decadal SST anomalies with the WTP multidecadal variability and AMO in observations and ATL_VARMIX simulation. (a) Time series of the decadal SST anomalies averaged over the SNP region (20°N–35°N, 170°E–155°W) in the ERSST, HadSST3 and ATL_VARMIX datasets. The time series of SST anomalies are linearly detrended, filtered using an 11-year running mean, and normalized to unit variance. (b) Cross correlations as a function of the time lag (year) of the SNP decadal SST anomalies in ERSST with the AMO (red) and WTP SST multidecadal variability (blue) for the period 1900–2013. Positive (negative) lags

indicate that the WTP SST/AMO (SNP SST) is leading. The dashed lines are the 95% confidence levels based on the effective numbers of degrees of freedom. (c) As in (b), but for the observed unfiltered time series of AMO, SNP and WTP SST anomalies, to further support the simultaneous relationship of the SNP SST with the AMO and WTP SST. (d) As in (b), but for the results in the ATL_VARMIX simulation.

2, Similarly, what are the cross correlations between SNP decadal SST anomalies and the AMO at all time leads/lags? i.e. are they in phase or there is some time lead/lag between them?

Response: Following the reviewer's comments, we also investigated the phase relationship between the AMO and SNP decadal SST anomalies. The cross correlations between the AMO and SNP decadal SST anomalies in both the simulation and the observation are shown in Fig. A1b and A1d (the red line) as a function of the time lag. In the simulations, the cross correlation between the AMO and SNP decadal SST peaks at lag zero with a high and significant correlation above 0.8. In the observation, the positive correlation between the decadal SNP SST anomalies and the AMO is not only simultaneous but is also significant and slightly larger when the AMO leads by up to several years. Although the cross correlation indicates a small time lag (4 years) of the SNP SST relative to the AMO, the time scale of the lag is much shorter than the multidecadal time scale dominated in the two indices. This implies that it is still reasonable to regard them as approximately in-phase. Meanwhile, the simultaneous correlation between the AMO and SNP decadal SST anomalies is 0.85, indicating that a large fraction of the multidecadal variability in the SNP SST can be explained by the simultaneous variations of the AMO.

Similar to the analysis of WTP SST anomalies, we also analyzed the cross correlation using the unfiltered data of AMO and SNP SST anomalies in the observation (Fig. A1c). The simultaneous correlation at lag zero is strong ($r = 0.55$) and statistically significant. This positive correlation is also significant and slightly larger ($r = 0.58$) at lag 1 when the AMO leads the SNP SST by 1 year. At negative lags when the SNP SST leads the AMO, the correlation drops off quickly with time lag. This further supports the forcing of the SNP SST by the AMO. According to the reviewer's comments, we include the results for the relationship between AMO and SNP decadal SST anomalies in the revised manuscript (Supplementary Fig. 9 and lines 301-310).

The above cross correlation analyses between the AMO, SNP and WTP decadal SST anomalies (the responses to the comments #1 and #2 from the reviewer) suggest that during the AMO warm (cold) phase, there is significant SST warming (cooling) occurring over the SNP and WTP regions. This in-phase relationship is reproduced by the Atlantic Pacemaker experiment (ATL_VARMIX). In the manuscript, we proposed a physical mechanism (as shown in the schematic diagram in the manuscript) and

provided both dynamical and modelling evidence to suggest that the SNP decadal SST anomalies act as a key bridge linking the AMO and WTP SST. This “SNP bridge” mechanism reasonably well explains the observed and simulated relationship between the AMO, SNP and WTP SST anomalies. In order to further confirm that the AMO influences the WTP SST through the SNP bridge, we carried out an additional Atlantic Pacemaker experiment, which is similar to the ATL_VARMIX, but with prescribed climatological SST over the SNP region (as the reviewer pointed out in the comment #3 below). The results from this experiment suggest that the AMO has no clear influence on the WTP SST without the SNP bridge (Please see the response to the comments #3 below for details), further supporting the “SNP bridge” mechanism.

3, Since SNP SST anomalies are identified as a key bridge linking the AMO and WTP SST anomalies, if possible it would be nice to have an experiment as ATL_VARMIX but with prescribed climatology SST over the SNP region, i.e. if there are no SNO SST anomalies, can WTP decadal SST anomalies still be influenced by the AMO ?

Response: We appreciate the good comments from the review, which help to increase the reliability of our conclusion. Following the reviewer’s suggestion, we carried out an additional Atlantic Pacemaker experiment. This experiment is similar to the ATL_VARMIX, but with prescribed climatological SST over the SNP region. This experiment aims to investigate whether the WTP decadal SST anomalies can still be influenced by the AMO without the SNP SST feedback, and is referred to as ATL_VARMIX_SNPCLIM.

The simulated Pacific response (SST and atmospheric circulation) to the AMO forcing in the ATL_VARMIX_SNPCLIM simulation is shown in Fig. A2. Compared with the ATL_VARMIX experiment, the correlations of simulated WTP decadal SST anomalies with the AMO become rather weak and insignificant (Fig. A2a, correlation decreases from 0.90 to 0.41, with explained variance reduced by 64%), and the amplitude of AMO-related decadal SST variability over the WTP region is considerably reduced (Fig. A2b, amplitude decreases from 0.09 K to 0.024 K, reduced by 73%). In addition, without the SNP SST feedback, the Pacific atmospheric circulation response is also different from the result in the ATL_VARMIX. The significant anomalous low and cyclonic flow over the WTP region are absent in the ATL_VARMIX_SNPCLIM, and particularly, the converging flow from the tropics toward the SNP region also disappears, due to the exclusion of SNP SST feedback. This indicates that, without the feedback of the SNP SST anomalies, the AMO is unlikely to have a significant impact on the WTP decadal variability. Therefore, these results support our conclusion that the SNP SST feedback is very important for the remote influence of AMO on the WTP. These results have been included in the revised manuscript (Supplementary Fig. 10 and lines 340-351)

Figure A2 Simulated Pacific response to the AMO in ATL_VARMIX_SNPCLIM simulation (see Methods for experiment description). (a) Correlation map between Pacific–Atlantic SST and the AMO at decadal time scales. Dots indicate the correlations significant at the 95 % confidence level. (b) Regression map of Pacific–Atlantic SST (units: K) on the normalized AMO index at decadal time scales. (c) Regressions of SLP (shading, units: hPa) and surface winds (vectors, units: m s^{-1} ; omitted below 0.1 m s^{-1}) on the normalized AMO index at decadal time scales. In (b) and (c), dots indicate the regressions significant at the 95 % confidence level. The long-term trends in all variables were removed prior to the correlation and regression analyses.

4, A very recent observational based analysis (Barcikowska et al. 2017) on the teleconnection of multidecadal variability also shows that warmer WTP SST and negative SLP anomalies occur during the positive AMO phase, and vice versa (Fig. 2 A,C), and discussed that “Changes of SSTs in the tropical and extratropical North Atlantic are mirrored by changes throughout the western Pacific and Indian Oceans”. Fig. 4 in Barcikowska et al. 2017 shows the regression of the observed velocity potential at 850hPa and 200hPa on the AMO, and discussed that “Warming observed in the tropical North Atlantic is physically consistent with the anomalous large-scale low-level convergence and high-level divergence tendencies, indicative of strengthened convection in this region. Cooling in the tropical central Pacific is associated with a relative subsidence tendency or suppressed convection tendencies in that region”, quite

consistent with the proposed mechanism (Fig. 3 and Fig. 5) in the manuscript. It would be very useful to cite this study to support the conclusion of this manuscript.

Barcikowska, M.J., T. R. Knutson, R. Zhang, 2017, *Observed and Simulated Fingerprints of Multidecadal Climate Variability and Their Contributions to Periods of Global SST Stagnation*, *Journal of Climate*, 30, 721-737, doi: <http://dx.doi.org/10.1175/JCLI-D-16-0443.1>

Response: Thanks for the good comments from the reviewer. We agree with the reviewer. The results presented in the latest paper (Barcikowska et al. 2017) about the observed response of atmospheric circulation to the AMO are consistent with the findings in our manuscript. The observed anomalous low over the WTP region, the low-level convergence and upper-level divergence over the North Atlantic and the upper-level convergence over the central-eastern Pacific during the positive AMO phase are similar to those found in the Atlantic Pacemaker experiment. Thus, these can support the conclusion as an observational evidence. Therefore, following the reviewer's suggestion, we cite the study in the revised manuscript (please see lines 253 and 258).

5, *Supplementary Fig. 1: it would be nice to show the cross-correlation between low-pass filtered WTP SST and IPO/PDO at other time lags as well, so the time lead/lags at the maximum correlations can be seen.*

Response: We thank the reviewer for the good comments. Following the reviewer's suggestion, we show the cross-correlation between low-pass filtered WTP SST and IPO/PDO in the revised manuscript. As shown in Fig. A3, the maximum correlations can be seen at a time lag around -10, with the WTP SST leading by about 10 year. The simultaneous correlations are near zero, and no significant correlations are found when the IPO/PDO leads the WTP SST. Figure A3 has been included in the revised manuscript as Supplementary Fig. 1b.

Figure A3 Cross correlations as a function of the time lag (year) of the low-pass filtered WTP SST with the IPO (red) and PDO (blue) indices. Positive (negative) lags indicate

that the IPO/PDO (WTP SST) is leading. The dashed lines are the 95% confidence levels based on the effective numbers of degrees of freedom.

6, Supplementary Fig. 4 and 5 are very important results, and I would suggest to move them to the main manuscript. In particular, Supplementary Fig. 5 shows the spatial WTP SST pattern associated with the AMO, and is very useful to understand the discussions in the main text.

Response: Thanks for the good comments from the reviewer. We agree with the reviewer that Supplementary Fig. 4 and 5 are very important results. Following the reviewer's suggestion, these two figures have been moved to the main manuscript, and the figure numbers and figure references are changed accordingly.

7, Why the color bar ranges in Fig. 2f and Supplementary Fig. 7b are very different? Both are regressions of cloud cover on the decadal WTP SST index.

Response: We thank the reviewer for pointing this out. This is because the units for the cloud cover data derived from ICOADS dataset (original Supplementary Fig. 7b) are oktas, while the units for the simulated cloud cover are percentages (original Fig. 2f). Each okta represents one eighth of the sky (1 okta = 12.5%) covered by cloud, so the color bar range for the simulated cloud cover (in %) is about 10 times larger than that for the ICOADS cloud cover data (in oktas). According to the reviewer's comments, in the revised manuscript, we convert the ICOADS cloud amount data in oktas to percentages and use the same color bar range in the both figures (please see Supplementary Fig. 6 in the revised version).

8, Line 196, suggest changing the word "driving" to "amplifying", since here it refers to a positive feedback, not a driving mechanism.

Response: Following the reviewer's suggestion, the word has been changed in the revised manuscript (please see line 195).

II. Response to Comments of Reviewer #2

General comments:

The study investigates the mechanisms behind the western tropical Pacific (WTP) SST multidecadal variability. The manuscript claims that the multidecadal variability of the WTP is driven by the Atlantic Multidecadal Oscillation (AMO). The teleconnection process occurs in two steps:

- 1. the positive phase of the AMO generates upward motion over the North Atlantic that descends over the remote North Pacific, weakening the Aleutian low and reducing wind speed in the subtropical North Pacific westerlies. The weakening of the westerly winds in turn lead to a wind-evaporation-SST effect that warms the SST locally.*
- 2. The subtropical North Pacific SST warming induces wind convergence from the tropics, which drives an anomalous cyclonic circulation over the WTP. The anomalous atmospheric circulation eventually generates SST warming over the region via a SST - sea level pressure – cloud – longwave radiation feedback.*

The mechanism is very complex, but the authors provided enough evidence of the results. The findings are interesting, novel and worth publication.

Response: We thank the reviewer for the helpful comments and suggestions. We have revised the manuscript seriously and carefully, according to the reviewer's comments and suggestions. The point-to-point responses to the comments are listed as follows.

Comments:

One thing that the authors could have considered in this study is the use of a fully-dynamical ocean model, instead of a partially coupled mixed ocean model. This is because the simulation suggests a link between the AMO and the eastern Pacific as shown by the strong and significant correlations in Supplementary Fig.5d. Thus, there could exist a dynamical link across the ocean basin. Having said that, I understand that performing a Pacemaker experiment in such configuration may not be feasible for this study. But I wonder if this mechanism could be simulated in another model (e.g. pre-industrial control run from any cmip model that can reproduce a reasonable AMO). The authors correctly acknowledged the model design limitation at the end of the manuscript.

Response: We appreciate the helpful comments from the reviewer, which help to further verify the mechanisms proposed in this study. In light of the reviewer's comments, we performed additional analyses using the preindustrial control simulations from the CMIP5 models. We examined the relationship between the AMO and WTP SST in all available CMIP5 model simulations, and found two models that can reasonably reproduce the teleconnection between the AMO and WTP SST. One is the coupled model version 3 from GFDL (referred to as GFDL-CM3) and the other one is the CM5 from CNRM (referred to as CNRM-CM5). The control runs of the GFDL-

CM3 and the CNRM-CM5 are 500- and 800-year-long simulations, respectively, and the last 300 years of the control runs are analyzed. This choice may avoid the initial adjustment period because of the spin-up process in a fully coupled model. In addition, all the model data are linearly detrended to remove the possible remaining drift.

Figure B1 shows the results from the simulations of the two models. As displayed in Figure B1a, the time series of the decadal WTP SST, SNP SST and the AMO from the GFDL-CM3 model are characterized by evident multidecadal variations, and there is clearly an in-phase relationship between the three SST anomaly time series. The correlations between the three decadal indices are $R_{(AMO, WTP)} = 0.63$, $R_{(AMO, SNP)} = 0.52$ and $R_{(SNP, WTP)} = 0.60$, all significant at the 95% confidence level. This in-phase relationship is consistent with that revealed in the observation and Atlantic Pacemaker simulation (ATL_VARMIX). Figure B1b further shows the spatial patterns of decadal SST, SLP and surface winds in association with the AMO. Associated with the AMO warm SST anomaly, an anomalous low is seen over the North Atlantic, and over the Pacific, the SLP field is characterized by low-pressure (high-pressure) anomalies over Northwest Pacific and the WTP region (Northeast Pacific and eastern Pacific). Significant warm SST anomalies are seen over the WTP and SNP regions. Due to the SNP SST warming, the surface wind field shows a strong converging flow from the tropics towards the SNP region, leading to an anomalous cyclonic flow over the WTP region, which is in accordance with the anomalous low there. These features of the Pacific response to the AMO are generally consistent with those revealed in the Atlantic Pacemaker simulation. Moreover, in the GFDL-CM3 simulation, there is clear evidence that the AMO-related SNP SST warming strongly influences the Pacific atmospheric circulation, providing a favourable condition for the development of WTP SST warming. Therefore, it can also be identified as a key bridge linking the AMO and WTP SST multidecadal variations, consistent with the proposed mechanism. The results from the CNRM-CM5 simulation are largely similar to the GFDL-CM3 (Fig. B1c and B1d), nevertheless in the CNRM-CM5, the responses over the Pacific to the AMO seem stronger. Although the above results from the two CMIP5 models are preliminary, they provide clear evidence that the proposed mechanisms in the present study may be simulated in other models, such as a fully coupled GCM.

The present study focuses on the teleconnection between the AMO and WTP multidecadal variability. This teleconnection is successfully reproduced in the coupled slab ocean model (ATL_VARMIX simulation), and we use the simulation to further investigate the possible physical mechanisms involved. As the reviewer also pointed out, the employment of slab ocean model is a model design limitation of the present study, and thus we provide a discussion on this limitation at the end of the manuscript. Nevertheless, the above results from the CMIP5 models provide additional confidence that the proposed mechanisms may be simulated in the fully coupled GCM. Therefore, following the reviewer's suggestion, we plan to conduct further systematic and in-depth investigations using the fully coupled GCM in a forthcoming paper.

Figure B1 (a) Simulated time series of the decadal WTP SST, SNP SST and the AMO in the preindustrial control run of GFDL-CM3. The time series are filtered using an 11-year running mean and normalized to unit variance. (b) Regression maps of decadal anomalies of SST (shading, units: K), SLP (contours, units: Pa) and surface winds (vectors, units: m s^{-1} ; omitted below 0.1 m s^{-1}) on the normalized AMO index in the GFDL-CM3 simulation. Dots indicate the SST regressions significant at the 95% confidence level. (c)–(d) As in (a)–(b), but for the simulation from the CNRM-CM5 preindustrial control run. The black and green boxes indicate the WTP and SNP regions, respectively.

Line 199 seems to have the wrong figure reference. Should that be Figure 2e or Supplementary Figure 6b?

Response: We thank the reviewer for pointing this out. The figure reference is corrected in the revised manuscript (please see line 198).

Model description: The ICTPAGCM is coupled to a slab ocean model in the Indo-Pacific region and monthly varying SSTs in the Atlantic. The model design needs a clearer description, i.e. what is the forcing in the extratropical regions (e.g. southern latitudes)? How is treated the transition between slab ocean model and prescribed SST?

Response: We appreciate the important comments from the reviewer. Following the reviewer's suggestion, we provide a clearer description for the model design. In the slab ocean model, the mixed-layer is used everywhere in the Indo-Pacific region, including both tropics and extratropics, and in the sea-ice region, the surface temperature anomalies are calculated using a slab layer over ice. The transition between the slab ocean model and prescribed SSTs is accomplished by using a buffer zone with a spatial range of 7.5° , in which the SST is calculated from the weighted average of the modelled and prescribed SSTs. The weighting values are one in the prescribed-SST domain and

linearly reduced to zero in the buffer zone. These descriptions have been included in the revised manuscript (please see lines 447-453).

Figure 1b: for comparison purposes it may be better to scale the y-axis to the same as the other subpanels.

Response: We thank the reviewer for pointing this out. In the revised manuscript, the y-axis of Figure 1b is scaled to the same as the other subpanels.

Suppl. Fig. 3b: as above, it may look better if y-axis range is increased.

Response: The y-axis of Supplementary Fig. 3b is increased to have the same scale as Supplementary Fig. 3a.

III. Response to Comments of Reviewer #3

General comments:

Review's comments for the paper (NCOMMS-17-00941), entitled "Western tropical Pacific multidecadal variability forced by the Atlantic multidecadal oscillation" by Sun and co-authors

Recommendation: Major revision

This paper investigate the multidecadal sea surface temperature (SST) variability over western tropical Pacific (WTP). By using both observational data and numerical experiments, the paper suggests that the WTP multidecadal variability is largely explained by the remote Atlantic multidecadal oscillation (AMO). The paper further argues that the SST–sea level pressure (SLP)–cloud–longwave radiation positive feedback is the main physical process involved in the response of WTP SST change to AMO change. Results are interesting and they are generally well presented. Therefore, the paper is worth of publication. However, the paper needs some improvements by properly address the comments listed below before it can be accepted for publication in the Nature Communications.

Response: We would like to thank the reviewer for the constructive review comments/suggestions, which give us a big help to improve the quality of our manuscript. We have done our best to address the reviewer's concerns and modified the manuscript in the light of the reviewer's suggestions. Point-by-point responses to the reviewer's comments are listed below.

Major comments

1. Paper suggests that SST–sea level pressure (SLP)–cloud–longwave radiation positive feedback is the main physical process involved in the WTP SST response to AMO. With SST warming, it is expecting an increase in atmospheric water vapor through local enhanced evaporation. What is the role of water vapor feedback? The reviewer strongly recommends authors to investigate this issue, and quantify its role, at least in the pacemaker experiments.

Response: Thanks for the important comments from the viewer. We agree with the reviewer that it is very necessary to investigate and quantify the role of water vapor feedback in the WTP multidecadal variability. Following the reviewer's comments, we performed additional analyses to address this issue.

We investigated the effect of water vapor (WV) feedback on the WTP multidecadal variability using the ATL_VARMIX simulations. Figure C1a shows the time series of decadal variations of column-integrated WV, convective cloud cover and net surface longwave radiation (SLR) averaged over the WTP region. At decadal timescales, both cloud cover and column-integrated WV are highly and positively correlated with the net SLR over the WTP region (0.97 and 0.91 for cloud cover and WV, respectively, Fig. C1a). To extract the respective influence of cloud cover and WV, we further performed the partial correlation analysis. The partial correlation between the decadal SLR and WV is considerably reduced and becomes insignificant ($r = 0.35$) with the influence of cloud cover excluded, whereas that between the decadal SLR and cloud cover remains strong and significant ($r = 0.89$) after removing the contribution from the WV. Thus, over the WTP region, the decadal variability of SLR is more closely related to the cloud cover than to the WV, and the cloud feedback to the SLR is independent of the WV feedback. Moreover, we established an empirical linear model for the decadal SLR based on the cloud cover and column-integrated WV (See Fig. C1b), and the decadal SLR fitted using the linear model closely follows the simulated SLR ($r = 0.98$). The respective contribution from cloud feedback and WV feedback to the decadal SLR anomalies over the WTP region can be further estimated by using the linear model. In association with the WTP SST multidecadal variability, the contribution from WV feedback to the SLR anomalies is about 0.06 W m^{-2} (Fig. C1d), much smaller than that from the cloud feedback (0.31 W m^{-2} , Fig. C1e). Therefore, in the ATL_VARMIX simulation, the anomalous SLR associated with the WTP multidecadal variability is mainly controlled by the cloud feedback, whereas the WV feedback plays a minor role, although it also positively contributes to the SLR anomaly.

The atmospheric longwave heating rate (LHR) is also provided by the output of the ATL_VARMIX simulations. Positive (negative) LHR anomalies indicate a decrease (increase) in the atmospheric longwave radiative cooling, corresponding to more (less) longwave radiation trapped within the atmosphere. Thus, we further investigate the role of WV feedback by using the LHR field. We first examine the vertical distribution of the WV anomalies in association with the WTP multidecadal variability (Fig. C2a and C2b). Associated with the WTP SST warming, the atmospheric WV over the WTP region shows an increase at both the low level and upper level, and the WV increase is much stronger at the low level than at the upper level. If the WV feedback plays a dominant role, the LHR anomalies are expected to have larger amplitudes at the low level due to the greenhouse effect of WV. However, the vertical distribution of decadal LHR anomalies is inconsistent with that expected from the WV feedback (Fig. C2c and C2d). The low-level LHR anomalies associated with the WTP multidecadal variability are positive but rather weak over the WTP region, whereas at the upper level, the positive LHR anomalies are much stronger. This vertical distribution of decadal LHR anomalies is more consistent with the cloud feedback. The cloud top height of the deep convective cloud is high (above 500 hPa), and thus the increase of convective cloud amount associated with WTP SST warming can lead to a strong reduction of

atmospheric longwave radiative cooling (positive LHR anomalies) at upper levels beneath the cloud top. Therefore, the analysis of the vertical distribution of atmospheric LHR is consistent with the multiple regression analysis and further supports that the cloud feedback is the dominant process involved in the WTP multidecadal variability.

The present study focuses on the teleconnection between the AMO and WTP multidecadal variability. The results from the Atlantic Pacemaker simulation provides additional evidence for the teleconnection and thus support the main findings from the observation data. We use the simulation to investigate the possible physical mechanisms involved. A more detailed comparison of various feedback processes over the WTP region is not the emphasis of the present study. In the future, however, we will follow the reviewer's suggestions and conduct further systematic and in-depth investigations focusing on the various feedback processes. According to the reviewer's suggestion, the results for the multiple regression analysis (quantification of the role of WV feedback) are included in the revised manuscript (please also see Supplementary Fig. 5 and lines 200-221).

Figure C1 Effects of cloud feedback and water vapor (WV) feedback on the decadal net surface longwave radiation (SLR) anomalies in the ATL_VARMIX simulations. (a) Time series of the simulated decadal net SLR, convective cloud cover and column-integrated WV anomalies averaged over the WTP region. The time series are filtered using an 11-year running mean, and all smoothed indices are scaled to unit variance. (b) The simulated decadal net SLR anomalies (units: $W m^{-2}$) averaged over the WTP region and the linear model fit of the SLR based on the decadal cloud cover and column-integrated WV anomalies. The linear model is developed as: $SLR = a \times Cloud + b \times WV$, where the coefficients $a = 0.77 W m^{-2}/\%$ and $b = 0.21 W m^{-2}/(kg m^{-2})$ are determined empirically by multiple linear regression based on the simulation data over

the period of 1900–2013, so that the regression error of the linear model is minimized. (c) Regression map of column-integrated WV (units: kg m^{-2}) on the normalized WTP SST index at decadal time scales. Dots indicate the regressions significant at the 95 % confidence level. (d) Contribution from WV feedback to the decadal SLR anomalies (units: W m^{-2}) in association with the WTP multidecadal variability, as estimated by using the linear model (multiplying the column-integrated WV anomalies in (c) by the coefficient b). (e) As in (d), but for the contribution from cloud feedback, as estimated by multiplying the convective cloud cover anomalies in Fig. 4f by the coefficient a . (f) The total contribution from cloud feedback and WV feedback to the decadal SLR anomalies obtained by adding (d) to (e).

Figure C2 Analyses of the WV and longwave heating rate (LHR) fields. (a) Regression map of near surface WV (units: g kg^{-1}) on the normalized WTP SST index at decadal time scales. Dots indicate the regressions significant at the 95 % confidence level. (b) As in (a), but for the WV at 500 hPa. (c) As in (a), but for the decadal LHR anomalies (units: K/day) near surface. (d) As in (c), but for the LHR at 500 hPa.

2. *The study proposes that the change in SST over subtropical North Pacific (SNP) bridges the influence of AMO on the WTP multidecadal variability. However, in both observations and model experiments, the time series of SST variability over SNP is not analysed. To make your arguments more convincing, this time series needs to be analysed and its phase relationships with both AMO and the WTP multidecadal variability should be discussed.*

Response: We appreciate the good comments from the reviewer. In light of the reviewer’s suggestions, we analysed the time series of the SNP decadal SST anomalies and investigated its phase relationships with AMO and the WTP multidecadal

variability in both the observation and ATL_VARMIX simulation.

The time series of the SNP decadal SST variations in the simulation and observation datasets are shown in Fig. C3a, which both display evident multidecadal fluctuations and vary strongly in phase with each other. This indicates that the observed SNP decadal SST variability is reasonably reproduced by the ATL_VARMIX simulation. The cross correlations between the SNP and WTP decadal SST anomalies in both the simulation and the observation are further shown in Fig. C3b and C3d (the blue line) as a function of the time lag. Since the results from the two SST observation datasets are similar, here we show the observational based analysis from the ERSST data. The cross correlations between the observed SNP and WTP decadal SST anomalies are generally consistent with the simulated results, showing a high and significant correlation peak around lag zero. Although the maximum correlation occurs at lag 1 for the observation, it is still reasonable to consider the simultaneous relationship since the multidecadal time scale dominated in the two indices is much broader than the lag. The simultaneous correlations are 0.85 and 0.86 for the observation and the simulation, respectively. This suggests that at the decadal timescales, the WTP and SNP SST anomalies are largely in phase: warm (cool) WTP SST anomalies are likely to occur when there is an anomalous SST warming (cooling) over the SNP region.

To further confirm the simultaneous relationship between the WTP and SNP SST anomalies in the observation, we also analyzed the cross correlation using the unfiltered data. This analysis can also be used to test whether the 1-year lag really exists in the annually resolved time series. As shown in Fig. C3c, the maximum correlation appears at lag zero with a clear correlation peak, and the correlations quickly diminish with time lag (the 1-year lag vanishes). Therefore, this provides further evidence that the WTP and SNP SST anomalies vary largely in phase with each other.

The cross correlations between the AMO and SNP decadal SST anomalies in both the simulation and the observation are shown in Fig. C3b and C3d (the red line) as a function of the time lag. In the simulations, the cross correlation between the AMO and SNP decadal SST peaks at lag zero with a high and significant correlation above 0.8. In the observation, the positive correlation between the decadal SNP SST anomalies and the AMO is not only simultaneous but is also significant and slightly larger when the AMO leads by up to several years. Although the cross correlation indicates a small time lag (4 years) of the SNP SST relative to the AMO, the time scale of the lag is much shorter than the multidecadal time scale dominated in the two indices. This implies that it is still reasonable to regard them as approximately in-phase. Meanwhile, the simultaneous correlation between the AMO and SNP decadal SST anomalies is 0.85, indicating that a large fraction of the multidecadal variability in the SNP SST can be explained by the simultaneous variations of the AMO.

Similar to the analysis of WTP SST anomalies, we also analyzed the cross correlation using the unfiltered data of AMO and SNP SST anomalies in the observation

(Fig. C3c). The simultaneous correlation at lag zero is strong ($r = 0.55$) and statistically significant. This positive correlation is also significant and slightly larger ($r = 0.58$) at lag 1 when the AMO leads the SNP SST by 1 year. At negative lags when the SNP SST leads the AMO, the correlation drops off quickly with time lag. This further supports the forcing of the SNP SST by the AMO.

Figure C3 Phase relationship of the SNP decadal SST anomalies with the WTP multidecadal variability and AMO in observations and ATL_VARMIX simulation. (a) Time series of the decadal SST anomalies averaged over the SNP region (20°N – 35°N , 170°E – 155°W) in the ERSST, HadSST3 and ATL_VARMIX datasets. The time series of SST anomalies are linearly detrended, filtered using an 11-year running mean, and normalized to unit variance. (b) Cross correlations as a function of the time lag (year) of the SNP decadal SST anomalies in ERSST with the AMO (red) and WTP SST multidecadal variability (blue) for the period 1900–2013. Positive (negative) lags indicate that the WTP SST/AMO (SNP SST) is leading. The dashed lines are the 95% confidence levels based on the effective numbers of degrees of freedom. (c) As in (b), but for the observed unfiltered time series of AMO, SNP and WTP SST anomalies, to further support the simultaneous relationship of the SNP SST with the AMO and WTP SST. (d) As in (b), but for the results in the ATL_VARMIX simulation.

The above cross correlation analyses between the AMO, SNP and WTP decadal SST anomalies suggest that during the AMO warm (cold) phase, there is significant SST warming (cooling) occurring over the SNP and WTP regions. This in-phase relationship is reproduced by the Atlantic Pacemaker experiment (ATL_VARMIX). In the manuscript, we proposed a physical mechanism (as shown in the schematic diagram in the manuscript) and provided both dynamical and modelling evidence to suggest that the SNP decadal SST anomalies act as a key bridge linking the AMO and WTP SST. This “SNP bridge” mechanism reasonably well explains the observed and simulated relationship between the AMO, SNP and WTP SST anomalies. In order to further

confirm that the AMO influences the WTP SST through the SNP bridge, we carried out an additional Atlantic Pacemaker experiment, which is similar to the ATL_VARMIX, but with prescribed climatological SST over the SNP region. This experiment aims to investigate whether the WTP decadal SST anomalies can still be influenced by the AMO without the SNP SST feedback, and is referred to as ATL_VARMIX_SNPCLIM.

The simulated Pacific response (SST and atmospheric circulation) to the AMO forcing in the ATL_VARMIX_SNPCLIM simulation is shown in Fig. C4. Compared with the ATL_VARMIX experiment, the correlations of simulated WTP decadal SST anomalies with the AMO become rather weak and insignificant (Fig. C4a, correlation decreases from 0.90 to 0.41, with explained variance reduced by 64%), and the amplitude of AMO-related decadal SST variability over the WTP region is considerably reduced (Fig. C4b, amplitude decreases from 0.09 K to 0.024 K, reduced by 73%). In addition, without the SNP SST feedback, the Pacific atmospheric circulation response is also different from the result in the ATL_VARMIX. The significant anomalous low and cyclonic flow over the WTP region are absent in the ATL_VARMIX_SNPCLIM, and particularly, the converging flow from the tropics toward the SNP region also disappears, due to the exclusion of SNP SST feedback. This indicates that, without the feedback of the SNP SST anomalies, the AMO is unlikely to have a significant impact on the WTP decadal variability. Therefore, these results further support the “SNP bridge” mechanism that the SNP SST feedback is very important for the remote influence of AMO on the WTP.

According to the reviewer’s suggestion, we include the results and the discussion for the relationship between AMO, SNP and WTP decadal SST anomalies in the revised manuscript (please see Supplementary Fig. 9 and lines 301-310), and the analysis of the results from the ATL_VARMIX_SNPCLIM experiment is also included (please see Supplementary Fig. 10 and lines 340-351).

Figure C4 Simulated Pacific response to the AMO in ATL_VARMIX_SNPCLIM simulation (see Methods for experiment description). (a) Correlation map between Pacific–Atlantic SST and the AMO at decadal time scales. Dots indicate the correlations significant at the 95 % confidence level. (b) Regression map of Pacific–Atlantic SST (units: K) on the normalized AMO index at decadal time scales. (c) Regressions of SLP (shading, units: hPa) and surface winds (vectors, units: m s^{-1} ; omitted below 0.1 m s^{-1}) on the normalized AMO index at decadal time scales. In (b) and (c), dots indicate the regressions significant at the 95 % confidence level. The long-term trends in all variables were removed prior to the correlation and regression analyses.

3. *Conclusion and discussion could be more concise by focusing on main points. There is no need to repeat some statements that are already in results section.*

Response: We thank the reviewer for the good comments. According to the reviewer’s suggestion, we revise the conclusion, remove the repeated statements and make it more concise by focusing on the main findings: “The subtropical North Pacific (SNP) SST anomalies are identified as a key bridge linking the AMO and WTP SST multidecadal fluctuations. The AMO SST anomalies induce an atmospheric teleconnection from North Atlantic into North Pacific, influencing the SNP SST through the wind–evaporation–SST effect. The AMO-induced SNP SST anomalies provide a feedback to

the Pacific atmospheric circulation, which eventually generates SST anomalies over the WTP region via a SST–sea level pressure–cloud–longwave radiation feedback.” (please also see 369-375 lines in the revised version).

Minor comments

1. *Lines 62-65. Need to make a distinction between simultaneous relationship in winter and delayed impact of ENSO on summer climate over East Asia.*

Response: We thank the reviewer for pointing this out. Following the reviewer’s suggestion, we revise the introduction of the WTP interannual SST variability: “At interannual time scales, there is a strong simultaneous relationship between the WTP SST and ENSO in winter, and the WTP SST anomalies play an important role in conveying the delayed impact of ENSO on summer climate over East Asia.” (please also see lines 61-64 in the revised version).

2. *Lines 93-95. There are many studies about AMO remote impacts, but the authors only cite their own two papers published very recently. Suggest to cite some of those early studies.*

Response: Thanks for the important comments. Following the reviewer’s suggestion, in the revised manuscript, we cite two early studies about the remote impacts of AMO on Indian monsoon (Goswami et al. 2006), East Asian summer monsoon (Lu et al. 2006) and ENSO variance (Dong et al. 2006)

References:

Goswami, B. N., Madhusoodanan, M. S., Neema, C. P. & Sengupta, D. A physical mechanism for North Atlantic SST influence on the Indian summer monsoon. *Geophys. Res. Lett.* **33**, L02706 (2006).

Lu, R. Y., Dong, B. W. & Ding, H. Impact of the Atlantic multidecadal oscillation on the Asian summer monsoon. *Geophys. Res. Lett.* **33**, L24701 (2006).

Dong, B. W., Sutton, R. T. & Scaife, A. A. Multidecadal modulation of El Nino-Southern Oscillation (ENSO) variance by Atlantic Ocean sea surface temperatures. *Geophys. Res. Lett.* **33**, L08705 (2006)

3. *Lines 118-121. These power spectra are based on about 140 year data. Need some comments on limitation.*

Response: Thanks for the important comments. We agree with the reviewer that due to the shortness of observational record length, the SST anomaly time series are not long

enough to be able to make a very strong statement on the multidecadal periodicity. In this study, we focus on the multidecadal variability of the WTP SST rather than its potential multidecadal periodicity. Thus, following the reviewer's suggestion, we add some comments on the limitation in the revised manuscript: "... quasi-60 years (here the shortness of observational record length forms a limitation in the spectral analysis), indicating pronounced multidecadal variability over the WTP region." (please also see lines 120-122 in the revised version).

4. *Supplementary figure 5d indicates pattern of negative phase of IPO. What is the relationship between AMO and IPO (PDO) indices in the model experiment? Does IPO in the model experiments play a role in the WTP multidecadal variability?*

Response: We thank the reviewer for the good comments. Following the reviewer's comments, we performed additional analyses to investigate the phase relationship of the IPO with the AMO and WTP multidecadal variability in the ATL_VARMIX simulation. We first calculated the IPO index in the model experiment. The IPO index is defined as the difference between the SST anomaly averaged over the central equatorial Pacific (10°S–10°N, 170°E–90°W) and the average of the SST anomaly in the extratropical North (25°N–45°N, 140°E–145°W) and South Pacific (50°S–15°S, 150°E–160°W). This definition of IPO index is the same as that in Henley et al. (2015), which is known as a tripole index for the IPO (Fig. C5b). Figure C5a shows the correlations between the simulated IPO index and global SST at decadal timescales. The correlation map exhibits a tripole pattern across the Pacific, in accordance with the definition of the IPO index. However, over the WTP region and the Atlantic basin, the correlations are rather weak and statistically insignificant. We also calculated the cross correlations of the decadal IPO with the AMO and WTP decadal SST (Fig. C5c). The simultaneous correlations of the IPO with the AMO and WTP SST are weak and insignificant (-0.41 for the AMO and -0.45 for the WTP SST), and no significant correlations are found at other time leads/lags. The observational based analysis also shows a very weak simultaneous correlation between the IPO and WTP decadal SST anomalies. Thus, the simulation and observation are consistent with each other, and both indicate that the WTP multidecadal variability is unlikely to be explained by the IPO.

Reference:

Henley, B.J., Gergis, J., Karoly, D.J., Power, S.B., Kennedy, J., Folland, C.K., (2015). A Tripole Index for the Interdecadal Pacific Oscillation. *Clim. Dyn.* <http://dx.doi.org/10.1007/s00382-015-2525-1>

Figure C5 (a) Correlation map between global SST anomalies and the IPO at decadal timescales in the ATL_VARMIX simulation. Dots indicate the correlation significant at the 95% confidence level. (b) Time series of the decadal WTP SST, IPO and AMO in the ATL_VARMIX simulation. The time series are filtered using an 11-year running mean and normalized to unit variance. (c) Cross correlations as a function of the time lag (year) of the IPO with the AMO (blue) and WTP SST multidecadal variability (red) in the ATL_VARMIX simulation. Positive (negative) lags indicate that the IPO (AMO/WTP) is leading. The dashed lines are the 95% confidence levels based on the effective numbers of degrees of freedom.

5. *Lines 290-295. The information about these sensitive experiments should be in the method section.*

Response: Thanks for the good comments from the reviewer. The description of the sensitive experiments is moved to the Method section in the revised manuscript (please see lines 465-479).

REVIEWERS' COMMENTS:

Reviewer #1 (Remarks to the Author):

All my points raised in the previous round of review have been satisfactorily addressed. The revised manuscript has been substantially improved. I recommend the acceptance for publication of the revised manuscript in Nature Communications.

Reviewer #2 (Remarks to the Author):

The authors addressed/clarified the points I raised previously. The manuscript has improved from previous version and I recommend acceptance subject to minor revision as follow.

The authors assessed pre-industrial control runs from all available CMIP5 models and only two of them showed a reasonable teleconnection pattern between the AMO and WTP SST. In my opinion, this indicates the mechanism may not be robust across fully-coupled models. Detailed analyses are needed to explain why fully-coupled models do not reproduce this pattern. The authors plan to further investigate this in a separate study. I suggest including a sentence in the manuscript discussing the preliminary analysis using cmip models (i.e., two out of X models reproduce the mechanisms suggested in the paper).

Reviewer #3 (Remarks to the Author):

Review's comments for the revised paper (NCOMMS-17-00941A), entitled "Western tropical Pacific multidecadal variability forced by the Atlantic multidecadal oscillation" by Sun and co-authors

Recommendation: Accept

My comments on the early version of the paper have been addressed in a satisfactorily manner in the revised version. The paper is, therefore, acceptable for publication.

Responses to Reviewers (Reviewer comments in italics; Responses in standard case)

REVIEWERS' COMMENTS:

Reviewer #1 (Remarks to the Author):

All my points raised in the previous round of review have been satisfactorily addressed. The revised manuscript has been substantially improved. I recommend the acceptance for publication of the revised manuscript in Nature Communications.

Response: We thank the Reviewer once more for taking the time to provide constructive and helpful comments on the initial and revised versions of this paper.

Reviewer #2 (Remarks to the Author):

The authors addressed/clarified the points I raised previously. The manuscript has improved from previous version and I recommend acceptance subject to minor revision as follow.

Response: We thank the Reviewer once more for taking the time to provide constructive and helpful comments on the initial and revised versions of this paper.

The authors assessed pre-industrial control runs from all available CMIP5 models and only two of them showed a reasonable teleconnection pattern between the AMO and WTP SST. In my opinion, this indicates the mechanism may not be robust across fully-coupled models. Detailed analyses are needed to explain why fully-coupled models do not reproduce this pattern. The authors plan to further investigate this in a separate study. I suggest including a sentence in the manuscript discussing the preliminary analysis using cmip models (i.e., two out of X models reproduce the mechanisms suggested in the paper).

Response: We appreciate the good suggestion from the reviewer. We fully agree with the reviewer that further investigations using the CMIP5 fully coupled models are needed. Based on this point, a future study will be designed for comprehensive intercomparison and evaluation of the simulations from the CMIP5 models. Following the reviewer's suggestion, we include a sentence in the manuscript discussing the preliminary analysis using the CMIP5 models: "We performed a preliminary analysis using the preindustrial control simulations from the CMIP5 fully coupled models, and two out of the 18 analyzed models (providing a simulation of at least 500 years in length) reproduce the mechanisms suggested in this study." (lines 404-407 in the revised version).

Reviewer #3 (Remarks to the Author):

Review's comments for the revised paper (NCOMMS-17-00941A), entitled "Western tropical Pacific multidecadal variability forced by the Atlantic multidecadal

oscillation" by Sun and co-authors

Recommendation: Accept

My comments on the early version of the paper have been addressed in a satisfactory manner in the revised version. The paper is, therefore, acceptable for publication.

Response: We thank the Reviewer once more for taking the time to provide constructive and helpful comments on the initial and revised versions of this paper.